# Two distinguishable impurities in BEC: squeezing and entanglement of two Bose polarons

**Christos Charalambous[1*], Miguel A. Garcia-March[1], Aniello Lampo[1], Mohammad Mehboudi[1] and Maciej Lewenstein[1,2]**

**1** ICFO – Institut de Ciències Fotòniques, The Barcelona Institute of Science and Technology, 08860 Castelldefels (Barcelona), Spain
**2** ICREA, Lluis Companys 23, E-08010 Barcelona, Spain

⋆ christos.charalambous@icfo.eu

## Abstract

We study entanglement and squeezing of two uncoupled impurities immersed in a Bose-Einstein condensate. We treat them as two quantum Brownian particles interacting with a bath composed of the Bogoliubov modes of the condensate. The Langevin-like quantum stochastic equations derived exhibit memory effects. We study two scenarios: (i) In the absence of an external potential, we observe sudden death of entanglement; (ii) In the presence of an external harmonic potential, entanglement survives even at the asymptotic time limit. Our study considers experimentally tunable parameters.



## Contents

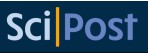

# 1   Introduction

Entanglement represents a necessary resource for a number of protocols in quantum information and for other quantum technologies, which are expected to be implemented in the foreseeable future for various practical applications [1–5]. The entangled parties of a composite quantum system evolve, under realistic conditions, coupled to external degrees of freedom, which may be treated as a *bath*. In this *open quantum system*, the bath acts as a source of decoherence, leading to the destruction of quantum coherence among the states of the entangled subsystems [6]. Indeed, to reach a relevant technological level, one of the main obstacles is the difficulty to ensure such a coherence despite the interaction of the system with the bath. However, the presence of the bath can produce other potentially useful phenomena. For instance, two non-interacting particles immersed in a common bath can be entangled as a consequence of an effective interaction induced by the bath [7,8]. A number of situations have been considered, where indeed entanglement is observed in the aforementioned setting [9–17]. Here we investigate the bath-induced entanglement among two distinguishable impurities embedded in a common homogeneous Bose-Einstein condensate (BEC) in 1D. Such an issue recently attracted a lot of attention, e.g, the study of impurities in double-wells in a BEC [18], the study of entanglement and the measurement of non-Markovianity between a pair of two-level localized impurities in a BEC [19,20], particle number entanglement between regions of space in a BEC [21], or the environment-induced interaction for impurities in a lattice [22,23]. Moreover, in a number of experiments [24–26] performed last year, entanglement between regions of a BEC was observed. In these works discrete observables were considered, while on the contrary, in our studies we will focus on entanglement of continuous variables in BEC.

The major motivation for us to undertake this work is the contemporary progress in the manipulation and control of ultracold atoms and ions, that paves the way to new possible experiments. The behaviour of an impurity in a Bose gas has recently attracted a lot of attention, both on the theoretical and experimental side [27–45]. The main feature of such a system lies in the creation of excitation modes (Bogoliubov quasiparticles) associated to the motion of the atoms of the gas, that dress the impurity, leading to the formation of a compound system named Bose polaron. For two such impurities within a BEC, studies have focused in the past on the possibility to form bound states (bipolarons) for sufficiently strong interactions between the impurities and the condensate atoms [46,47]. Furthermore, the Bose polaron problem was recently studied within the quantum Brownian motion (QBM) model [43,48], which describes the dynamics of a quantum particle interacting with a bath made up of a huge number of harmonic oscillators obeying the Bose-Einstein statistics [6,49–51]. In this analogy, the impurity plays the role of the Brownian particle and the Bogoliubov excitations of the BEC are the bath-oscillators. Here, by extending this view, we study the creation of entanglement between two different impurities in a Bose gas, as a consequence of the coupling induced by the presence of the Bogoliubov modes, which play the role of the bath.

Note that the QBM model for the motion of the two kinds of impurities in a BEC is a continuous-variable description *i.e.* it is expressed in terms of position and momentum operators. Thus, entanglement measures based on the density matrices are not conveniently calculable because the density matrix in this case is infinite-dimensional. We therefore use the logarithmic negativity [52] as a more fitting choice in this context. This measure is expressed in terms of the covariance matrix, namely a matrix, whose elements are all of the position and momentum related correlation functions.

To compute these correlation functions, we solve the Heisenberg equations of the system, which can be reduced to a quantum stochastic Langevin equation for each particle. This set of two coupled equations are non-local in time, namely the dynamics of both impurities in a BEC carry certain amount of memory. In this context, such a feature is often related to the super-Ohmic character of the spectral density, constituting the main quantity that embodies the properties of the bath. The presence of memory effects (non-Markovianity) can also be shown to lead to the appearance of entanglement [53–57]. The role of the memory effects in the works above is to preserve entanglement in the long-time regime [58] and in the high temperature regime [13]. Nevertheless, in these works the spectral density was assumed to be Ohmic, such that the non-Markovianity is purely attributed to the influence of one particle on the other. Indeed, the disturbances caused by particles to one another is mediated through the common bath, which take a finite time to propagate through the medium, making the evolution of each particle history-dependent. This results on a decay of entanglement in several stages [13,17] or to a limiting distance for bath induced two-mode entanglement [54]. In [59], the scenario of an additional source of non-Markovianity, emerging from a non-Ohmic spectral density was considered. The non-ohmic spectral density resulted in more robust entanglement among the two impurities.

In this work we study entanglement as a function of the physical quantities of the system, such as temperature, impurity-gas coupling, gas interatomic interaction and density. These parameters may be tuned in experiments allowing to control the amount of entanglement between the impurities. We distinguish the situation in which the impurity is trapped in a harmonic potential and that where it is free of any trap. In the trapped case, we also study squeezing which is a resource for quantum sensing.

The manuscript is organized as follows. In Sec. 2 we present the Hamiltonian of the system, showing that it can be reduced to that of two quantum Brownian particles interacting with a common bath of Bogoliubov modes. We also write the quantum Langevin equations, find the expression for the spectral density showing that it presents a super-ohmic form, and solve the equations in order to evaluate the position and momentum variances. Finally, we review and discuss the logarithmic negativity as an entanglement quantifier and a criterion which we use to detect two-mode squeezing. In Sec. 3.1 we study the out-of-equilibrium dynamics of untrapped impurities, while in Sec. 3.2 we study entanglement and squeezing of the two impurities as a function of the system parameters for the trapped case. In Sec. 4 we offer the conclusions and outlook. We discuss details on the derivations of the spectral density and susceptibility in appendices A and B. In App. C, we comment on the difficulty of finding an analytic solution for the trapped case even when the centers for the particles potentials coincide. Finally, in App. D we study, for the trapped case, the effective equilibrium Hamiltonian of the system reached at long-times.

## 2 The model system

### 2.1 Hamiltonian

We consider two kinds of distinguishable impurities of mass $m_1$ and $m_2$, immersed in a bath of interacting bosons of mass $m_B$ enclosed in a box of volume $V$. This system is described by the Hamiltonian

$$H = H_I^{(1)} + H_I^{(2)} + H_B + H_{BB} + H_{IB}^{(1)} + H_{IB}^{(2)}, \tag{1}$$

with

$$H_I^{(j)} = \frac{\mathbf{p}_j^2}{2m_j} + T_j\left(\mathbf{x}_j, \mathbf{d}_j\right), \tag{2a}$$

$$H_B = \int d^d\mathbf{x}\Psi^\dagger(\mathbf{x})\left(\frac{\mathbf{p}_B^2}{2m_B} + U(\mathbf{x})\right)\Psi(\mathbf{x}) = \sum_k \epsilon_k a_k^\dagger a_k, \tag{2b}$$

$$H_{BB} = g_B \int d^d\mathbf{x}\Psi^\dagger(\mathbf{x})\Psi^\dagger(\mathbf{x})\Psi(\mathbf{x})\Psi(\mathbf{x}) = \frac{1}{2V}\sum_{q,k',k} C_B(q)a_{k'-q}^\dagger a_{k+q}^\dagger a_{k'}a_k, \tag{2c}$$

$$H_{IB}^{(j)} = \frac{1}{V}\sum_{q,k} C_{IB}^{(j)}(k)\rho_I^{(j)}(q)a_{k-q}^\dagger a_k, \tag{2d}$$

with $a^\dagger$ and $a$ being the bath creation and annihilation operators respectively, $\mathbf{x}_j$ and $\mathbf{p}_j$ the position and momentum operators of particle $j = 1,2$, where they satisfy the commutation relations $[a^\dagger, a] = 1$ and $[\mathbf{x}_j, \mathbf{p}_k] = i\hbar\delta_{jk}$. Here, we consider the bosons to be in a homogeneous medium, i.e. $U(\mathbf{x}) = $ const, and the external potential experienced by the impurities is

$$T_j\left(\mathbf{x}_j, \mathbf{d}_j\right) = \sum_{i=1}^3 \frac{m_j\mathbf{\Omega}_j^2(\mathbf{x}_j + \mathbf{d}_j)^2}{2}, \tag{3}$$

that is a 3D parabolic potential centered at $\mathbf{d}_j = (d_{j,x}, d_{j,y}, d_{j,z})$ and with frequency $\mathbf{\Omega}_j = (\Omega_{j,x}, \Omega_{j,y}, \Omega_{j,z})$. We will consider both the trapped $\Omega_{j,i} > 0$ and untrapped cases $\Omega_{j,i} = 0$ where $i = x, y, z$. The densities of the impurities in the momentum domain $\rho_I^{(j)}(q)$ are

$$\rho_I^{(j)}(q) = \int_{-\infty}^{\infty} e^{-iq\mathbf{x}}\delta\left(\mathbf{x} - \left(\mathbf{x}_j + \mathbf{d}_j\right)\right)d^d\mathbf{x}. \tag{4}$$

The Fourier transforms of the interactions among the $j^{th}$ impurity and the bath and among bath particles themselves are, respectively

$$C_{IB}^{(j)}(k) = \mathcal{F}_{IB}\left[g_{IB}^{(j)}\delta\left(\mathbf{x} - \mathbf{x}'\right)\right], \tag{5a}$$

$$C_B(q) = \mathcal{F}_B\left[g_B\delta\left(\mathbf{x} - \mathbf{x}'\right)\right]. \tag{5b}$$

The coupling constants in Eq. (5) are

$$g_{IB}^{(j)} = 2\pi\hbar^2\frac{a_{IB}^{(j)}}{m_R^{(j)}}, \tag{6a}$$

$$g_B = 4\pi\hbar^2\frac{a_B}{m_B}, \tag{6b}$$

where $m_R^{(j)} = \frac{m_B m_j}{(m_B + m_j)}$, is the reduced mass, and $a_{IB}^{(j)}$ and $a_B$ are the scattering lengths between the impurities and the bath particles and between the bath particles themselves, respectively.

By performing a Bogoliubov transformation,

$$a_{\mathbf{k}} = u_{\mathbf{k}} b_{\mathbf{k}} - w_{\mathbf{k}} b_{-\mathbf{k}}^{\dagger} \, , \; a_{\mathbf{k}}^{\dagger} = u_{\mathbf{k}} b_{-\mathbf{k}} - w_{\mathbf{k}} b_{\mathbf{k}}^{\dagger}, \tag{7}$$

with

$$u_{\mathbf{k}}^2 = \frac{1}{2}\left( \frac{\epsilon_{\mathbf{k}} + n_0 C_{\mathrm{B}}}{E_{\mathbf{k}}} + 1 \right),$$

and

$$w_{\mathbf{k}}^2 = \frac{1}{2}\left( \frac{\epsilon_{\mathbf{k}} + n_0 C_{\mathrm{B}}}{E_{\mathbf{k}}} - 1 \right),$$

the terms related purely with the bath particles transform to the following non-interacting term

$$H_{\mathrm{B}} + H_{\mathrm{BB}} = \sum_{\mathbf{k} \neq 0} E_{\mathbf{k}} b_{\mathbf{k}}^{\dagger} b_{\mathbf{k}}, \tag{8}$$

where we neglected some constant terms, such that the diagonal form of the Hamiltonian is only valid up to quadratic order in the bath operators. In the above expression, $E_{\mathbf{k}} = \hbar c |\mathbf{k}| \sqrt{1 + \frac{1}{2}(\xi \mathbf{k})^2} \equiv \hbar \omega_{\mathbf{k}}$ is the Bogoliubov spectrum, where

$$\xi = \frac{\hbar}{\sqrt{2 g_{\mathrm{B}} m_{\mathrm{B}} n_0}} \tag{9}$$

is the coherence length for the BEC and

$$c = \sqrt{\frac{g_{\mathrm{B}} n_0}{m_{\mathrm{B}}}} \tag{10}$$

is the speed of sound in the BEC. Finally, $n_0$ is the density of the bath particles in the ground state, which turns out to be constant as a result of the homogeneity of the BEC. For a macroscopically occupied condensate, i.e. $N_{k \neq 0} \ll N_0$ where $N_k$ is the occupation number of the $k^{th}$ state, we obtain the following expression for the interaction part of the Hamiltonian

$$H_{\mathrm{IB}}^{(j)} = n_0 C_{\mathrm{IB}}^{(j)} + \sqrt{\frac{n_0}{V}} \sum_{\mathbf{k} \neq 0} \rho_{\mathrm{I}}^{(j)}(\mathbf{k}) C_{\mathrm{IB}}^{(j)} \left( a_{\mathbf{k}} + a_{-\mathbf{k}}^{\dagger} \right), \tag{11}$$

which after the Bogoliubov transformation, becomes

$$H_{\mathrm{IB}}^{(j)} = \sum_{\mathbf{k} \neq 0} V_{\mathbf{k}}^{(j)} e^{i\mathbf{k} \cdot (\mathbf{x}_j + \mathbf{d}_j)} \left( b_{\mathbf{k}} + b_{-\mathbf{k}}^{\dagger} \right). \tag{12}$$

Here, the couplings are

$$V_{\mathbf{k}}^{(j)} = g_{\mathrm{IB}}^{(j)} \sqrt{\frac{n_0}{V}} \left[ \frac{(\xi \mathbf{k})^2}{(\xi \mathbf{k})^2 + 2} \right]^{\frac{1}{4}}. \tag{13}$$

After this procedure the Hamiltonian is transformed into

$$H = H_{\mathrm{I}}^{(1)} + H_{\mathrm{I}}^{(2)} + \sum_{\mathbf{k} \neq 0} E_{\mathbf{k}} b_{\mathbf{k}}^{\dagger} b_{\mathbf{k}} + \sum_{\mathbf{k} \neq 0} \left[ V_{\mathbf{k}}^{(1)} e^{i\mathbf{k} \cdot (\mathbf{x}_1 + \mathbf{d}_1)} + V_{\mathbf{k}}^{(2)} e^{i\mathbf{k} \cdot (\mathbf{x}_2 + \mathbf{d}_2)} \right] \left( b_{\mathbf{k}} + b_{-\mathbf{k}}^{\dagger} \right). \tag{14}$$

The above Hamiltonian with a non-linear interacting part is in general difficult to treat, and requires the usage of influence functional techniques [8]. To simplify the situation we will resort to the so called long-wave or dipole approximation. This is expressed by the assumption

$$\mathbf{k} \cdot \mathbf{x}_1 \ll 1, \; \mathbf{k} \cdot \mathbf{x}_2 \ll 1, \tag{15}$$

such that we can approximate the exponentials by a linear function. The validity of the linear approximation was studied in [43]. For the untrapped impurities, it turned into a condition of a maximum time as a function of $T$ for which the linear approximation holds. For the trapped case, the condition was over the trapping frequency as a function of $T$. The results we present here fulfil these conditions. With this, the Hamiltonian reads

$$H_{\text{Lin}} = H_{\text{I}}^{(1)} + H_{\text{I}}^{(2)} + \sum_{\mathbf{k} \neq 0} E_{\mathbf{k}} b_{\mathbf{k}}^{\dagger} b_{\mathbf{k}} + \sum_{\mathbf{k} \neq 0} \widetilde{V}_{\mathbf{k}} \left[ \mathbb{I} + i \mathbf{k} f_{\mathbf{k}} (\mathbf{x}_1, \mathbf{x}_2, \mathbf{d}_1, \mathbf{d}_2) \right] \left( b_{\mathbf{k}} + b_{-\mathbf{k}}^{\dagger} \right),$$

with $\mathbb{I}$ the identity and where

$$\widetilde{V}_{\mathbf{k}} (\mathbf{d}_1, \mathbf{d}_2) := V_{\mathbf{k}}^{(1)} e^{i \mathbf{k} \cdot \mathbf{d}_1} + V_{\mathbf{k}}^{(2)} e^{i \mathbf{k} \cdot \mathbf{d}_2}, \tag{16}$$

and

$$f_{\mathbf{k}} (\mathbf{x}_1, \mathbf{x}_2, \mathbf{d}_1, \mathbf{d}_2) := \frac{V_{\mathbf{k}}^{(1)} \mathbf{x}_1 e^{i \mathbf{k} \cdot \mathbf{d}_1} + V_{\mathbf{k}}^{(2)} \mathbf{x}_2 e^{i \mathbf{k} \cdot \mathbf{d}_2}}{V_{\mathbf{k}}^{(1)} e^{i \mathbf{k} \cdot \mathbf{d}_1} + V_{\mathbf{k}}^{(2)} e^{i \mathbf{k} \cdot \mathbf{d}_2}}. \tag{17}$$

Performing the transformation $b_{\mathbf{k}} \rightarrow b_{\mathbf{k}} - \frac{\widetilde{V}_{-\mathbf{k}} (\mathbf{d}_1, \mathbf{d}_2)}{E_{\mathbf{k}}} \mathbb{I}$, one obtains the following Hamiltonian

$$H_{\text{Lin}} = \sum_{j=1,2} H_{\text{I}}^{(j)} + i \sum_{\substack{j=1,2 \\ \mathbf{k} \neq 0}} \hbar g_{\mathbf{k}}^{(j)} \left( e^{i \mathbf{k} \cdot \mathbf{d}_j} b_{\mathbf{k}} - e^{-i \mathbf{k} \cdot \mathbf{d}_j} b_{\mathbf{k}}^{\dagger} \right) \mathbf{x}_j + \sum_{\mathbf{k} \neq 0} E_{\mathbf{k}} b_{\mathbf{k}}^{\dagger} b_{\mathbf{k}} + W (\mathbf{d}_1, \mathbf{d}_2) [\mathbf{x}_1 + \mathbf{x}_2], \tag{18}$$

with

$$W (\mathbf{d}_1, \mathbf{d}_2) = 2i \sum_{\mathbf{k} \neq 0} \frac{\mathbf{k} V_{\mathbf{k}}^{(1)} V_{\mathbf{k}}^{(2)} \cos (\mathbf{k} \cdot \mathbf{R}_{12})}{\hbar \omega_{\mathbf{k}}}, \tag{19}$$

where $\mathbf{R}_{jq} = |\mathbf{d}_j - \mathbf{d}_q|$, and $g_{\mathbf{k}}^{(j)} = \frac{\mathbf{k} V_{\mathbf{k}}^{(j)}}{\hbar}$. At this point we note that, to preserve the bare oscillator potential in Eq. (8) and hence ensure a positively defined Hamiltonian, it is conventional to introduce a counter term to the Hamiltonian. In this way the Hamiltonian at hand can be interpreted as a minimal coupling theory with $U(1)$ gauge symmetry [60]. This term is important because its introduction guarantees that no "runaway" solutions appear in the system, as shown in [61]. Nevertheless, we are committed not to introduce any artificial terms in the Hamiltonian and maintaining the fact that we are considering a Hamiltonian that describes a physical system. We will however identify a condition under which the problem of "runaway" solutions will not appear.

For the trapped impurities case, Hamiltonian (18) shows similarities to the Hamiltonian that describes the interaction of three harmonic oscillators in a common heat bath [59]. However, the two Hamiltonians differ in the following three aspects: First, our Hamiltonian describes the interaction as a coupling between the position of the particle and a modified expression of the momentum of the bath particles while [59] describes a position-position interaction. Second, the term $W (\mathbf{d}_1, \mathbf{d}_2) [\mathbf{x}_1 + \mathbf{x}_2]$ is absent in the Hamiltonian in [59]; Finally, our Hamiltonian lacks the counter term which is artificially introduced in [59] that results in a renormalization of the potential of the harmonic oscillator.

## 2.2 Heisenberg equations

From here on we treat only the one dimensional case, i.e. we assume that the BEC and the impurities are so tightly trapped in two directions as to effectively freeze the dynamics in those directions. In practice, the one dimensional coupling constant has to be treated appropriately,

as discussed in [62]. To study the out-of-equilibrium dynamics of the system, we first obtain the Heisenberg equations, which read as

$$\frac{dx_j}{dt} = \frac{i}{\hbar}\left[H, x_j\right] = \frac{p_j}{m_j}, \tag{20a}$$

$$\frac{dp_j}{dt} = \frac{i}{\hbar}\left[H, p_j\right] = -m_j\Omega_j^2 x_j(t) - i\sum_{k\neq 0}\hbar g_k^{(j)}\left(b_k e^{ikd_j} - b_k^\dagger e^{-ikd_j}\right), \tag{20b}$$

$$\frac{db_k}{dt} = \frac{i}{\hbar}\left[H, b_k\right] = -i\omega_k b_k - \sum_{j=1}^{2} g_k^{(j)} e^{-ikd_j} x_j, \tag{20c}$$

$$\frac{db_k^\dagger}{dt} = \frac{i}{\hbar}\left[H, b_k^\dagger\right] = i\omega_k b_k^\dagger - \sum_{j=1}^{2} g_k^{(j)} e^{ikd_j} x_j. \tag{20d}$$

Next we solve the equations of motion for the bath particles (20c) and (20d),

$$b_k(t) = b_k(0) e^{-i\omega_k t} - \int_0^t \sum_{j=1}^{2} g_k^{(j)} e^{-ikd_j} x_j(s) e^{-i\omega_k(t-s)} ds, \tag{21}$$

$$b_k^\dagger(t) = b_k^\dagger(0) e^{i\omega_k t} - \int_0^t \sum_{j=1}^{2} g_k^{(j)} e^{ikd_j} x_j(s) e^{i\omega_k(t-s)} ds. \tag{22}$$

Replacing these in Eqs.(20a)-(20b) yields the following equations of motion for the two particles

$$m_j\ddot{x}_j + m_j\Omega_j^2 x_j + m_j W(d_1, d_2) - \int_0^t \sum_{q=1}^{2} \lambda_{jq}(t-s) x_q(s) ds = B_j(t, d_j), \tag{23}$$

where $B_j(t, d_j)$ plays the role of the stochastic fluctuating forces, given by

$$B_j(t, d_j) = \sum_{k\neq 0} i\hbar g_k^{(j)}\left[e^{i(\omega_k t - kd_j)} b_k^\dagger(0) - e^{-i(\omega_k t - kd_j)} b_k(0)\right], \tag{24}$$

and the memory friction kernel $\lambda_{jq}(t-s)$ reads as

$$\lambda_{jq}(t) = \sum_{k\neq 0}\Theta\left(t - \left|\frac{kR_{jq}}{\omega_k}\right|\right)\tilde{g}_k^{(jq)}\sin\left(kR_{jq} + \omega_k(t-s)\right), \tag{25}$$

with, $\tilde{g}_k^{(jq)} = 2\hbar g_k^{(j)} g_k^{(q)}$. In the literature, $\lambda_{jq}(t)$, is also refered to as the dissipation kernel, or also the susceptibility. The Heaviside function guarantees causality by introducing the corresponding retardation due to the finite distance $d_j - d_q$ between the centers of the two harmonic oscillators. The two are related by the Kubo formula as

$$\lambda_{jq}(t-s) = -i\Theta\left(t - \left|\frac{kR_{jq}}{\omega_k}\right|\right)\left\langle\left[B_j(t, d_j), B_q(s, d_q)\right]\right\rangle_{\rho_B}. \tag{26}$$

Hence, we understand that the reason the Heaviside step function arises is due to the fact that the forces commute for time-like separations. On the left hand side of Eq. (23) appears a restoring force which is originated by the fact that the impurities are similar to two harmonic oscillators. Furthermore, also non-local terms appear due to the interaction of the impurities with the environment, in particular a dissipative self-force and a history-dependent non-Markovian interaction between the two impurities. Both of these non-linear terms are a

consequence of the coupling of the impurities to the bath. On the right hand side, the stochastic fluctuating force appears with Gaussian statistics, i.e. the first moment, which is assumed to be $\langle B_j(t, d_j) \rangle = 0$ for $j = 1, 2$, and the second moment of the probability distribution related to the state of the stochastic driving force $B_j(t, d_j)$ is enough to describe the state of the bath.

There is another equivalent way of writing the equations of motion for the particles, in terms of the damping kernel $\Gamma_{jq}(t-s)$, related to the susceptibility as $\frac{1}{m_j}\lambda_{jq}(t-s) := -\frac{\partial}{\partial t}\Gamma_{jq}(t-s)$, which will enable us to identify a condition on the range of parameters for which our system is valid. Making use of the Leibniz integral rule, one can show that

$$-\frac{1}{m_j}\int_0^t \lambda_{jq}(t-s)x_q(s)\,ds = -\Gamma_{jq}(0)x_q(t) + \frac{\partial}{\partial t}\int_0^t \Gamma_{jq}(t-s)x_q(s)\,ds. \qquad (27)$$

With this, one can rewrite the equations of motion for each particle position in terms of the damping kernel

$$\ddot{x}_j + \Omega_j^2 x_j - \sum_{q=1}^2 \Gamma_{jq}(0)x_q(t) + \frac{1}{m_j}W(d_1, d_2) + \frac{\partial}{\partial t}\left[\int_0^t \sum_{q=1}^2 \Gamma_{jq}(t-s)x_q(s)\,ds\right] = \frac{1}{m_j}B_j(t, d_j). \qquad (28)$$

One can identify that $\Gamma_{jq}(0)x_q(t)$ in equations of motion (28) play the role of renormalization terms of the harmonic potential. Most importantly, these terms will not be present in case one includes a counter term in the initial Hamiltonian, Eq. (18). Under the assumption of weak coupling, one expects that these terms will not affect the long-time behaviour of the system, as explained in [49]. At the end of this section we obtain the necessary condition that ensures the positivity of the Hamiltonian.

It is useful to write Eqs. (23) as a single vectorial equation,

$$\underline{\ddot{X}}(t) + \underline{\underline{\Omega}}^2 \underline{X}(t) - \underline{\underline{M}}^{-1}\int_0^t \underline{\underline{\lambda}}(t-s)\underline{X}(s)\,ds = \underline{\underline{M}}^{-1}\left(\underline{B}^T(t, d_1, d_2) - \underline{W}(d_1, d_2)\mathbb{I}\right)\mathbb{I}, \qquad (29)$$

where

$$\underline{X}(t) = \begin{pmatrix} x_1(t) \\ x_2(t) \end{pmatrix}, \quad \underline{\underline{M}} = \begin{pmatrix} m_1 & 0 \\ 0 & m_2 \end{pmatrix}, \quad \underline{\underline{\Omega}}^2 = \begin{pmatrix} \Omega_1^2 & 0 \\ 0 & \Omega_2^2 \end{pmatrix}, \qquad (30)$$

$$\underline{W}(d_1, d_2) = \begin{pmatrix} W_1(d_1, d_2) \\ W_2(d_1, d_2) \end{pmatrix}, \qquad (31)$$

$$\underline{\underline{\lambda}}(t-s) = \begin{pmatrix} \lambda_{11}(t-s) & \lambda_{12}(t-s) \\ \lambda_{21}(t-s) & \lambda_{22}(t-s) \end{pmatrix}, \qquad (32)$$

$$\underline{B}(t, d_1, d_2) = \begin{pmatrix} B_1(t, d_1) \\ B_2(t, d_2) \end{pmatrix}. \qquad (33)$$

Equivalently, the vectorial equation that corresponds to Eqs. (28) (i.e. in terms of the damping kernel), is

$$\underline{\ddot{X}}(t) + \underline{\underline{\tilde{\Omega}}}^2 \underline{X}(t) + \frac{\partial}{\partial t}\int_0^t \underline{\underline{\Gamma}}(t-s)\underline{X}(s)\,ds = \underline{\underline{M}}^{-1}\left(\underline{B}^T(t, d_1, d_2) - \underline{W}(d_1, d_2)\right)\mathbb{I}, \qquad (34)$$

where

$$\underline{\underline{\Gamma}}(t-s) = \begin{pmatrix} \Gamma_{11}(t-s) & \Gamma_{12}(t-s) \\ \Gamma_{21}(t-s) & \Gamma_{22}(t-s) \end{pmatrix}, \qquad (35)$$

$$\underline{\tilde{\Omega}}^2 = \begin{pmatrix} \Omega_1^2 - \Gamma_{11}(0) & -\Gamma_{12}(0) \\ -\Gamma_{21}(0) & \Omega_2^2 - \Gamma_{22}(0) \end{pmatrix}. \tag{36}$$

We then introduce the transformation matrix $\underline{Q} = \underline{O}\,\underline{X}$ such that Eq. (34) transforms into

$$\ddot{\underline{Q}}(t) + \underline{\tilde{\Omega}_\mathrm{D}}^2 \underline{Q}(t) + \frac{\partial}{\partial t}\int_0^t \underline{\underline{\Xi}}(t-s)\underline{Q}(s)\,ds = \underline{\underline{M}}^{-1}\left(\underline{D}^T(t,d_1,d_2) - \underline{Q}W(d_1,d_2)\right)\mathbb{I}, \tag{37}$$

where $\underline{\underline{\Xi}}(t-s) = \underline{O}\,\underline{\underline{\Gamma}}(t-s)\underline{O}^T$, $\underline{D}^T(t,d_1,d_2) = \underline{O}\,\underline{B}^T(t,d_1,d_2)$ and

$$\underline{\underline{\tilde{\Omega}_\mathrm{D}}} = \underline{O}\,\underline{\tilde{\Omega}}^2 \underline{O}^T = \begin{pmatrix} \tilde{\Omega}_\mathrm{D}^1 & 0 \\ 0 & \tilde{\Omega}_\mathrm{D}^2 \end{pmatrix}, \tag{38}$$

i.e. it is diagonal. In the following sections we solve Eqs. (29) and (34) for the cases under study, by considering the Laplace or Fourier transforms as is presented in section 2.4.

We now identify from Equation (29) the condition under which even though the Hamiltonian lacks an *ad hoc* introduced renormalization term, this does not affect the long-time dynamics of the particles. This condition is that both $\tilde{\Omega}_\mathrm{D}^1, \tilde{\Omega}_\mathrm{D}^2 > 0$, because this way the Hamiltonian remains positive–definite and diverging solutions are avoided. In particular, it is required that

$$\frac{1}{2}\left[\Gamma_{11}(0) + \Gamma_{22}(0) - \Omega_1^2 - \Omega_2^2 + \left[4\Gamma_{12}(0)\Gamma_{21}(0) + \left(\Gamma_{11}(0) - \Gamma_{22}(0) - \Omega_1^2 + \Omega_2^2\right)^2\right]^{1/2}\right] < 0. \tag{39}$$

This imposes a restriction on the coupling constants range. Note that if we decouple the second particle i.e. if $g_k^{(2)} = 0$ for all $k$, then we obtain the same condition as for the one particle, $\Omega_1^2 > \Gamma_{11}(0)$ [43].

## 2.3 Spectral density

Let us write the dissipation kernel in Eq. (25) as

$$\lambda_{jq}(t-s) = \int_0^\infty \left[J_{jq}^\mathrm{antisym.}(\omega)\cos(\omega(t-s)) + J_{jq}^\mathrm{sym.}(\omega)\sin(\omega(t-s))\right]d\omega, \tag{40}$$

where we identify the spectral densities as

$$J_{jq}^\mathrm{antisym.}(\omega,t-s) = \sum_{k\neq 0}\Theta\left(t-s-\left|\frac{kR_{jq}}{\omega_k}\right|\right)\tilde{g}_k^{(jq)}\delta(\omega-\omega_k)\sin(kR_{jq}), \tag{41}$$

and

$$J_{jq}^\mathrm{sym.}(\omega,t-s) = \sum_{k\neq 0}\Theta\left(t-s-\left|\frac{kR_{jq}}{\omega_k}\right|\right)\tilde{g}_k^{(jq)}\delta(\omega-\omega_k)\cos(kR_{jq}). \tag{42}$$

We note that $g_k^{(j)}g_k^{(q)}$ is an even function of $k$, which implies that $J_{jq}^\mathrm{antisym.}(\omega) = 0$. Then, in App. A we show that in the continuum limit of the spectrum, Eq. (40) takes the following integral form for a system with 1D environment

$$\lambda_{jq}(t-s) = \int_0^\infty \Theta\left(t-s-\left|\frac{k_\omega R_{jq}}{\omega}\right|\right)J_{jq}(\omega)\sin(\omega(t-s))\,d\omega, \tag{43}$$

with

$$J_{jq}(\omega) = \widetilde{\tau}_{jq}\omega^3 \cos\left(k_\omega R_{jq}\right)\chi_{1D}(\omega),\tag{44}$$

where

$$\widetilde{\tau}_{jq} = 2\tilde{m}\tau\eta_j\eta_q,\tag{45}$$

with

$$\eta_j = \frac{g_{\mathrm{IB}}^{(j)}}{g_{\mathrm{B}}},\tag{46}$$

and

$$\tau = \frac{1}{2\pi\tilde{m}}\left(\frac{m_{\mathrm{B}}}{n_0 g_{\mathrm{B}}^{1/3}}\right)^{3/2},\tag{47}$$

represents the relaxation time, with $\tilde{m} = \frac{m_1 m_2}{m_1 + m_2}$. For the single impurity case, $R_{12} = 0$ and $\eta_2 = 0$, we obtain a cubic dependence of the spectral density on $\omega$, in accordance with the results obtained in [43]. It is worth noting here that the validity of the Fröhlich type Hamiltonian we have here imposes a restriction on $\eta_j$ [38,63], namely

$$\eta_j \lesssim \pi\sqrt{\frac{2n_0}{g_{\mathrm{B}}m_{\mathrm{B}}}}.\tag{48}$$

Therefore, we restrict ourselves within this limit, which for typical values of the related parameters, such as for example $g_{\mathrm{B}} = 2.36 \times 10^{-37} J \cdot m$ and $n_0 = 7\,(\mu m)^{-1}$ which are values that were experimentally considered in [34], becomes $\eta^{(cr)} \approx 7$. This condition is satisfied for all the values of $\eta_j$, $g_{\mathrm{B}}$ and $n_0$ considered here. Finally, $k_\omega$ in Eq. (44) is the inverse of the Bogoliubov spectrum

$$k_\omega = \frac{1}{\xi}\sqrt{\sqrt{1 + 2\left(\frac{\xi\omega}{c}\right)^2} - 1},$$

and the susceptibility is

$$\chi_{1D}(\omega) = 2\sqrt{2}\left(\frac{\Lambda}{\omega}\right)^3 \frac{\left[\sqrt{1 + \frac{\omega^2}{\Lambda^2}} - 1\right]^{\frac{3}{2}}}{\sqrt{1 + \frac{\omega^2}{\Lambda^2}}},\tag{49}$$

where $c$ is the speed of sound, $\xi$ the coherence length, and

$$\Lambda = \frac{g_{\mathrm{B}}n_0}{\hbar}.\tag{50}$$

One identifies two opposite limits for $\chi_{1D}(\omega)$, i.e. $\omega \ll \Lambda$ and $\omega \gg \Lambda$. Hence $\Lambda$ appears naturally as the characteristic cutoff frequency to distinguish between low and high frequencies. Note that the spectral density exhibits a super-ohmic behaviour given by the third power of the bath frequency in the continuous limit. Same behaviour was found in [64–66] for analogous problems, and it is attributed to the linear part of the Bogoliubov spectrum. With such a spectral density, it can be shown that certain quantities, e.g. the momentum dispersion would be divergent, unless the high frequencies are somehow removed from the spectrum. This can be achieved by introducing an ultraviolet cutoff given by $\Lambda$ such that only the part of the spectrum where $\omega < \Lambda$ remains. Upon doing so, $\chi_{1D}(\omega) \to 1$, and the Bogoliubov spectrum takes the linear form $\omega = c|k|$ such that the spectral density becomes

$$J_{jq}(\omega) = \widetilde{\tau}_{jq}\omega^3 \cos\left(\frac{\omega}{c}R_{jq}\right).\tag{51}$$

We consider two different possible analytical forms of the cut-off, the sharp one

$$J_{jq}(\omega) = \widetilde{\tau}_{jq}\omega^3 \cos\left(\frac{\omega}{c}R_{jq}\right)\Theta(\omega - \Lambda),$$ (52)

provided by an Heaviside function, and the exponential cutoff

$$J_{jq}(\omega) = \widetilde{\tau}_{jq}\omega^3 \cos\left(\frac{\omega}{c}R_{jq}\right)e^{-\frac{\omega}{\Lambda}}.$$ (53)

In [43], it is shown that the physics of the system in the long-time limit, *i.e.* associated to frequency regime $\omega \ll \Lambda$, does not depend on the existence, nor the form, of the cutoff. In this paper we will use both types of cutoffs depending on the problem at hand, namely we will use the exponential cutoff whenever we study the problem for distances between the traps of each kind of impurity different than 0 and the sharp cutoff otherwise. Finally, note that in comparison to the equations of motion obtained in [59], in our case the couplings of the two particles can be different, adding an extra parameter.

## 2.4 Solution of Heisenberg equations and covariance matrix

To evaluate the covariances one needs to solve the above equations of motion. In particular, we solve Eq. (34) by first obtaining the solution for the homogeneous equation, and then adding to that the particular solution [43,49],

$$\underline{X}(t) = \underline{\underline{G_1}}(t)\underline{X}(0) + \underline{\underline{G_2}}(t)\underline{\dot{X}}(0) + \int_0^t ds\,\underline{\underline{G_2}}(t-s)\left[\left(\underline{B}^T(s,d_1,d_2) - \underline{W}(d_1,d_2)\right)\mathbb{I}\right],$$ (54)

where

$$\mathcal{L}_z\left[\underline{\underline{G_1}}(t)\right] = \frac{z\mathbb{I} + \mathcal{L}_z\left[\underline{\underline{\Gamma}}(t)\right]}{z^2\mathbb{I} + \mathbb{I}\underline{\underline{\tilde{\Omega}}}^2 + z\mathcal{L}_z\left[\underline{\underline{\Gamma}}(t)\right]},$$ (55)

and

$$\mathcal{L}_z\left[\underline{\underline{G_2}}(t)\right] = \frac{1}{z^2\mathbb{I} + \mathbb{I}\underline{\underline{\tilde{\Omega}}}^2 + z\mathcal{L}_z\left[\underline{\underline{\Gamma}}(t)\right]},$$ (56)

with $\mathcal{L}_z[\cdot]$ denoting the Laplace transform. Notice that the second function is often referred to as the susceptibility, in analogy with the harmonic oscillator. It basically carries the same information as $\frac{\partial}{\partial t}\underline{\underline{\Gamma}}(t-s)$, which is what we refer to as susceptibility in this paper. We now need to evaluate the covariance matrix

$$C(0) = \begin{pmatrix} C_{\underline{XX}}(0) & C_{\underline{XP}}(0) \\ C_{\underline{PX}}(0) & C_{\underline{PP}}(0) \end{pmatrix},$$ (57)

with $\underline{P}(t) = (p_1(t), p_2(t))^T$ the momentum vector and

$$C_{AB}(t - t') = \frac{1}{2}\left\langle A(t)B^T(t') + B(t')A^T(t)\right\rangle.$$ (58)

Hence, the 4×4 matrix (57) is constructed from the vector $Y(t) = (x_1(t), x_2(t), p_1(t), p_2(t))$ as the product $\frac{1}{2}\langle\{\underline{Y}^T(t)\cdot\underline{Y}(t), (\underline{Y}^T(t)\cdot\underline{Y}(t))^T\}\rangle_{\rho_B}$. Furthermore, if we assume the initial state of the system to be of the Feynman-Vernon type, i.e. $\rho(0) = \rho_{12}(0)\otimes\rho_B$, then quantities like $\langle\underline{X}(0)\underline{B}^T(t, d_1, d_2)\rangle$ will vanish. We note that the appearance of the extra term $W(d_1, d_2)$ in the dynamics, absent in [59], indeed does not affect the evaluation of the covariance matrix since we are only interested in averages with respect to the state of the bath. In addition, the bath is assumed to be large enough such that the effects of the impurity dynamics on the

state of the bath are assumed to be negligible. Proceeding in a similar manner as in [43] for a thermal equilibrium bath at temperature $T$ we conclude that the equal time correlation function of position reads as

$$C_{x_j x_q}(0) = \hbar \int_0^\infty d\omega \coth\left[\frac{\hbar\omega}{2k_B T}\right] K_{jq}(\omega), \tag{59}$$

where

$$K_{jq}(\omega) = \sum_{k,s=1}^{2} \left(\mathcal{L}_{-i\omega}\left[\underline{\underline{G_2}}(t)\right]\right)_{jk} J_{ks}(\omega)\left(\mathcal{L}_{i\omega}\left[\underline{\underline{G_2}}(t)\right]\right)_{sq}. \tag{60}$$

Similarly, one can obtain the position-momentum and momentum-momentum blocks of the covariance matrix,

$$C_{x_j p_q}(0) = \hbar \int_0^\infty d\omega \left(im_q\omega\right)\coth\left[\frac{\hbar\omega}{2k_B T}\right] K_{jq}(\omega), \tag{61}$$

and

$$C_{p_j p_q}(0) = \hbar \int_0^\infty d\omega\, m_j m_q \omega^2 \coth\left[\frac{\hbar\omega}{2k_B T}\right] K_{jq}(\omega). \tag{62}$$

Notice that for $g_k^2 = 0\, \forall k$, i.e. by removing the second particle from the system, one reproduces the expressions provided in [43] for the single particle case, by considering the Laplace transform of the damping kernel $\mathcal{L}_z\left[\underline{\underline{\Gamma}}(t)\right]$. Importantly, in the presence of the second particle, the expressions for (59), (61) and (62) can still be obtained with the method described in [43], but only when $d_j - d_q = 0$, i.e. when the centers of the particle potentials are in the same position. Indeed, for $d_j - d_q = 0$, the Laplace transform of the damping kernel can be computed as

$$\mathcal{L}_z\left[\Gamma_{jq}(t)\right] = z\tilde{\tau}_{jq}\left[\Lambda - z\arctan\left(\frac{\Lambda}{z}\right)\right]. \tag{63}$$

However, if $d_j - d_q \neq 0$, $\mathcal{L}_z\left[\underline{\underline{\Gamma}}(t)\right]$ cannot be obtained as the integral does not converge. In such case, we use the method presented in [59], where the Fourier transform of the susceptibility is evaluated instead, through the usage of the Hilbert transform to solve Eq. (29). In the following, we find that we circumvent the aforementioned problem by using the Fourier method. By doing so, we are avoiding the integral on the imaginary axis, since the following relation is known between the Fourier and Laplace transforms

$$\tilde{f}_\pm(\omega) = \lim_{\epsilon \to 0}[\hat{f}(\epsilon + i\omega) \pm \hat{f}(\epsilon - i\omega)], \tag{64}$$

for even (+) and odd (-) functions respectively, for the functions $J(\omega)$, i.e. the spectral density, and $\lambda(\omega)$, i.e. the susceptibility, where the definition of the Fourier transform is [67]

$$\tilde{f}(\omega) = \int_{-\infty}^{+\infty} dt\, e^{-i\omega t} f(t)[\theta(t) + \theta(-t)]. \tag{65}$$

The correlation functions take the following form

$$C_{x_j x_q}(0) = \frac{\hbar}{2\pi}\int_{-\infty}^{\infty} d\omega \coth\left[\frac{\hbar\omega}{2k_B T}\right] Q_{jq}(\omega), \tag{66}$$

where

$$Q_{jq}(\omega) = \sum_{k,s=1}^{2} \left(\underline{\underline{\alpha}}(\omega)\right)_{js} \cdot \text{Im}\left[\underline{\underline{\lambda}}(\omega)\right]_{sk} \cdot \left(\underline{\underline{\alpha}}(-\omega)\right)_{kq}, \tag{67}$$

and Im[.] is the imaginary part. Here,

$$\underline{\underline{\alpha}}(\omega) = \frac{1}{-\omega^2 \mathbb{I} + \underline{\underline{\Omega}}^2 \mathbb{I} + \frac{1}{\underline{\underline{M}}} \mathcal{F}_\omega\left[\underline{\underline{\lambda}}(t)\right]}. \tag{68}$$

From [59] we have

$$\text{Im}[\underline{\underline{\lambda}}(\omega)] = -\hbar\left(\Theta(\omega) - \Theta(-\omega)\right)\underline{\underline{J}}(\omega), \tag{69}$$

where $\underline{\underline{\lambda}}(\omega)$ is the Fourier transform of the susceptibility $\underline{\underline{\lambda}}(t)$. We remind that the bath is assumed to be at thermal equilibrium. Equation (66) was proven in [68]. The other autocorrelation functions can be obtained in a similar way once we relate the momentum to the time derivative of the position as

$$C_{x_j p_q}(0) = \frac{\hbar}{2\pi}\int_{-\infty}^{\infty} d\omega\left(im_q\omega\right)\coth\left[\frac{\hbar\omega}{2k_B T}\right]Q_{jq}(\omega), \tag{70}$$

and

$$C_{p_j p_q}(0) = \frac{\hbar}{2\pi}\int_{-\infty}^{\infty} d\omega\, m_j m_q \omega^2 \coth\left[\frac{\hbar\omega}{2k_B T}\right]Q_{jq}(\omega). \tag{71}$$

To proceed further, one needs to evaluate $\mathcal{F}_\omega\left[\underline{\underline{\lambda}}(t)\right]$ that appears in Eq. (68). We already know the imaginary part of the susceptibility from Eq. (69). The real part of the susceptibility can be obtained by making use of the Kramers-Kronig relations, which mathematically means that one has to take the Hilbert transform $\mathcal{H}$ of the imaginary part of the susceptibility

$$\begin{aligned}
\text{Re}\left[\lambda_{jq}(\omega')\right] &= \mathcal{H}\left[\text{Im}[\lambda_{jq}(\omega)]\right](\omega') \\
&= \frac{1}{\pi}\mathcal{P}\int_{-\infty}^{\infty}\frac{\text{Im}[\lambda_{jq}(\omega)]}{\omega - \omega'}d\omega,
\end{aligned} \tag{72}$$

where Re is the real part and $P$ denotes the Cauchy principal value.

For the particular form of the spectral density in Eq. (53), the susceptibility takes the following form:

$$\begin{aligned}
\lambda_{jq}(\omega) = &-\frac{\hbar\tilde{\tau}_{jq}}{\pi}\left\{\omega^3\text{Re}\left[g(\omega) - g(-\omega)\right] + \pi\omega^3\text{Im}\left[\Theta(\omega)e^{-\frac{\omega}{\Lambda}+i\frac{\omega}{c}R_{jq}} + \Theta(-\omega)e^{\frac{\omega}{\Lambda}-i\frac{\omega}{c}R_{jq}}\right]\right. \\
&\left. + 2\omega^2\frac{\Lambda}{1+\left(\Lambda\frac{R_{jq}}{c}\right)^2} + 4\frac{\left(\frac{1}{\Lambda^3} - 3\frac{R_{jq}^2}{\Lambda c^2}\right)}{\left(\frac{1}{\Lambda^2} + \frac{R_{jq}^2}{c^2}\right)^3}\right\} - i\hbar\Theta(\omega)\tilde{\tau}_{jq}\omega^3\cos\left(\frac{\omega}{c}R_{jq}\right)e^{-\frac{\omega}{\Lambda}},
\end{aligned} \tag{73}$$

with

$$g(\omega) = e^{-\frac{\omega}{\Lambda}+i\frac{\omega}{c}R_{jq}}\Gamma\left[0, -\frac{\omega}{\Lambda} + i\frac{\omega}{c}R_{jq}\right], \tag{74}$$

where $\Gamma[\alpha, z] = \int_{-\infty}^{z} t^{\alpha-1}e^{-t}dt$ denotes the incomplete gamma function.

Finally, we remark here that for the case $R_{12} = 0$, one could proceed by using the Laplace transform as initially intended. With the work in [57] in mind, one could think that an analytic expression could be obtained, but we explain Appendix C, that this method is not applicable for a super-ohmic spectral density, and hence numerical integration should be applied instead, as is the case for the method using the Fourier transform.

### 2.4.1 Entanglement measure

We will address the existence and dependence of entanglement between the two impurities on the parameters of the model, both for trapped and untrapped impurities. For continuous-variable systems, an entanglement measure based on the density matrix is not conveniently calculable because the density matrix in this case is infinite-dimensional. For Gaussian bipartite states however, there are a number of ways to circumvent this problem [69]. In general, a state that is not separable is considered entangled. The well known Peres-Horodecki separability criterion [70,71], which poses a necessary condition for separability, was shown to be easily formulated for a Gaussian quantum bipartite state through the usage of the symplectic eigenvalues of the covariance matrix of the bipartite system. However, this criterion alone does not allow for a quantification of the entanglement in the system, because it does not offer a way to quantify entanglement that is a monotonic function of the aforementioned symplectic eigenvalues [72]. For pure states a unique quantification measure exists, which is the entropy of entanglement [73]. This is defined as the von Neumann entropy of the reduced states of the bipartite system. For mixed states however this is not the case. In this case, there are a number of possible measures of entanglement, such as the entanglement of formation [74], the distillable entanglement [75] and the logarithmic negativity [52]. The first two are notoriously hard to calculate in general. For this reason we resort to the usage of logarithmic negativity as the most convenient measure to quantify entanglement. The logarithmic negativity is defined as

$$E_{\text{LN}}(\rho_{12}) = \max[0, -\ln(2\nu_-)], \tag{75}$$

where $\rho_{12}$ is the density matrix of the two impurities and $\nu_-$ is the smallest symplectic eigenvalue of the partial transpose covariance matrix $C^{T_2}(0)$, where the partial transpose is taken with respect to the basis of only one of the impurities. Here, we briefly present the method to obtain the symplectic eigenvalues of $C^{T_2}(0)$. To do so, it will be more convenient to reconstruct the covariance matrix in Eq. (57) using the rearranged vector $\hat{Y}(t) = (x_1(t), p_1(t), x_2(t), p_2(t))$ such that the matrix becomes

$$C(0) = \begin{pmatrix} C_{11}(0) & C_{12}(0) \\ C_{21}(0) & C_{22}(0) \end{pmatrix}.$$

Thus, the diagonal matrices $C_{11}(0)$, $C_{22}(0)$ represent the covariance matrices of the first and second particle respectively, and $C_{21}(0) = C_{12}^T(0)$ represent the correlations between the two particles. It can be shown that, for such Gaussian quantum bipartite systems, partial transposition corresponds to time reversal [76]. Therefore the partial transpose of $C(0)$ with respect to the second particle degrees of freedom is obtained by assigning a minus sign to all the entries of the matrix where $p_2(t)$ appears. The symplectic spectrum then is obtained as the standard eigenspectrum of $\left| iQC^{T_2}(0) \right|$ where $Q$ is the symplectic form and $|.|$ represents taking the absolute value. The 4×4 symplectic matrix is defined as

$$\left[ \hat{Y}_j(t), \hat{Y}_q(t) \right] = i\hbar Q_{jq}, \tag{76}$$

which in our case it reads as

$$Q = \begin{pmatrix} 0 & 1 & 0 & 0 \\ -1 & 0 & 0 & 0 \\ 0 & 0 & 0 & 1 \\ 0 & 0 & -1 & 0 \end{pmatrix}.$$

The final result is a diagonal matrix $C_{\text{diag.}}^{T_2}(0)$ with diagonal entries $\text{diag}(\nu_1, \nu_1, \nu_2, \nu_2)$ from which the smallest one is selected and introduced in Eq. (75) to obtain a quantification of entanglement. At the Hilbert space level, this symplectic diagonalization transforms the state

into a tensor product of independent harmonic oscillators [52], each of which is in a thermal state, the temperature of which is a function of $\nu_j$. We recall that the Peres-Horodecki criterion states that if a density matrix $\rho_{12}$ of a bipartite system is separable, then $\rho_{12}^{T_2} \geq 0$. The logarithmic negativity quantifies how much this condition is not satisfied [69].

In [77,78] it was shown that logarithmic negativity quantifies the greatest amount of EPR correlations which can be created in a Gaussian state by means of local operations, and in [79] it was shown that logarithmic negativity provides an upper bound to distillable entanglement. The form of the symplectic eigenvalues can be explicitly given for the bipartite Gaussian system above using symplectic invariants constructed from determinants of the covariance matrix as [53,80]

$$\nu_{\pm} = \sqrt{\frac{\Delta \pm \sqrt{\Delta^2 - 4\det\left[C^{T_2}(0)\right]}}{2}}, \tag{77}$$

where $\Delta = \det\left[C_{11}(0)\right] + \det\left[C_{22}(0)\right] - 2\det\left[C_{12}(0)\right]$. In this scenario, the uncertainty relation can also be conveniently expressed in terms of the symplectic invariants constructed from determinants of the covariance matrix. In particular, it becomes equivalent to the following three conditions

$$C^{T_2}(0) > 0, \tag{78a}$$

$$\det\left[C^{T_2}(0)\right] \geq \frac{1}{2}, \tag{78b}$$

$$\Delta \leq 1 + \det\left[C^{T_2}(0)\right]. \tag{78c}$$

This can also be proven to be equivalent to the condition that the lowest eigenvalue of the symplectic matrix of $C(0)$, is larger than $\frac{1}{2}$, i.e.

$$\tilde{\nu}_{-} \geq \frac{1}{2}. \tag{79}$$

We notice here that the logarithmic negativity can take arbitrarily large values as $\nu_{-}$ can in principle go to 0, with the uncertainty principle still being satisfied. To attain maximal, finite entanglement, one has to fix the values of both local and global purities of the state of the two-impurity system [77,78]. Note also that one can construct an estimate of entanglement, the average logarithmic negativity, that is a function of these purities, which are easier to measure in an experiment [77,78]. However, in this work we focus on the study of the logarithmic negativity itself, because for the amount of entanglement that we find in our numerical results, average logarithmic negativity is not a good estimate, i.e. the error as defined in [77,78] is large.

We comment here on the experimental feasibility of our studies. From a practical point of view, there are two kind of terms that appear in the covariance matrix that one should evaluate, single particle expectation values, such as $\langle x_j^2 \rangle$, $\langle p_j^2 \rangle$, and crossterms such as $\langle x_1 x_2 \rangle$, $\langle p_1 p_2 \rangle$. For the former, there are already experiments in which one is able to evaluate them [34]. The idea is that one measures the position (or momentum) of the particle using a time-of-flight experiment in a system with a two species ultracold gas, in which one of the species is much more dilute - dilute enough as to consider its atoms as impurities immersed in a much bigger BEC. The position variance is obtained from the time-of-flight experiments, by releasing the atoms into free space initially and after allowing for the free expansion of their wavefunction for some time, measuring their position by irradiating them with a laser. Nevertheless, this method is not ideal for obtaining the real space information of a trapped sample, since during the free expansion process, signals from other atoms can easily be mixed with the signal of the atoms of interest. Furthermore, the current status of time-of-flight experiments does not allow

for the measurement of the crossterm covariances, which as of now there are no experiments to measure them, but we believe that one should be able in principle to do so.

In particular, a quantum gas microscope [81, 82] might be an option. This technique uses optical imaging systems to collect the fluorescence light of atoms, and has been used in the study of atoms in optical lattices, achieving much better spatial resolution [81, 82], and avoiding the aforementioned problem with time-of-flight experiments. With this technique, in principle, one should be able to measure all elements of the covariance matrix.

In the past quantum gas microscopes have been used to study spatial entanglement between itinerant particles, by means of quantum interference of many body twins, which enables the direct measurement of quantum purity [83]. In addition, there is an alternative way to study entanglement in continuous variable system, and that is by means of the average logarithmic negativity [77]. This quantity, can be related to the global and local purities of our system, which are measurable quantities. Nevertheless this is a more brute measure, since it is not estimated directly from the state of our system, and hence it may miss to detect entanglement for the actual state of our system.

Even if this is the case for average logarithmic negativity, with the knowledge of the existence of this measure and the technique of quantum gas microscopes, one could still consider this way of measuring entanglement as the worst case scenario. In general, it is known that to measure entanglement in a system of continuous variables is a difficult task, which is a problem that is not restricted to our case.

### 2.4.2 Squeezing

Once the covariance matrix is obtained, *squeezing* in the long time limit state of the system can also be rigorously studied. To this end one can use a set of criteria identified in [80]. As is noted in this work, squeezing in phase space in a system is a consequence of the appearance of non-compact terms in the Hamiltonian, i.e. terms that do not preserve the particle number, of the form

$$H \sim a_j^\dagger a_q^\dagger \pm a_j a_q. \tag{80}$$

This is indeed the form of the interacting part of the Hamiltonian in Eq. (18) once we write it in terms of creation and annihilation operators of the harmonic oscillators

$$x_j \sim \left( a_j^\dagger + a_j \right),$$
$$p_j \sim \left( a_j^\dagger - a_j \right),$$

and observe that in the Hamiltonian, terms of the form

$$a_j^\dagger b_q^\dagger \pm a_j b_q$$

appear. The criterion derived in [80] states that if

$$\nu_{min} \le \frac{1}{2}, \tag{81}$$

where $\nu_{min}$ is the smallest normal eigenvalue of the covariance matrix (not to be mistaken with the symplectic eigenvalue), then the state is said to be squeezed.

## 3 Results

Before presenting any results, we note that the following checks are made in order to guarantee that the assumptions presented in the previous section are valid. For all the parameters for which we present results here we check:

1. that the constraints for the validity of the linear approximation described in [43] hold; In particular, for the case of untrapped particles we check that the condition

$$\chi^{(Un)} := \frac{k_B T}{\hbar c} \sqrt{\frac{\hbar \tau \underline{\underline{M}}^{-1}}{2}} \frac{t\Lambda}{\alpha(\eta)} < 1 \tag{82}$$

is fullfilled. This implies an upper bound on the time we can study the impurites. For the case of trapped particles the condition reads as

$$\chi^{(Tr)} := \frac{k_B T}{\hbar c} \sqrt{\frac{2\underline{\underline{M}} \, \underline{\underline{\Omega}} \, C_{XX}(0)}{\hbar}} < 1; \tag{83}$$

2. that the condition Eq. (39) for the positivity of the Hamiltonian holds;

3. that the interactions are not so strong as to make invalid the initial Hamiltonian, as discussed in [38,63] (see also discussion in [43]); In particular the interaction strengths $\eta_j$ for $j \in 1,2$ have to satisfy

$$\eta_j < \eta_{crit} := \pi \sqrt{\frac{2n_0}{m_B g_B}}; \tag{84}$$

4. that the Heisenberg uncertainty principle condition Eq. (79) holds;

5. that the high temperature limit obeys equipartition theorem, meaning that at the limit of $T \to \infty$

$$\left\langle x_j^2 \right\rangle, \left\langle p_j^2 \right\rangle \propto T^{\frac{1}{2}},$$

for $j \in 1,2$. We used the fact that this latter condition was violated as an indication of problems with the numerical integrations that we performed.

### 3.1 Out-of-equilibrium dynamics and entanglement of the untrapped impurities

In this section, we study the case of untrapped impurities, $\Omega_1 = \Omega_2 = 0$. We restrict our studies to the low temperature regime, which is given by the condition $k_B T \ll \hbar \Lambda$. In the untrapped impurities case, the time dependent expressions for the Mean Square Displacement (MSD), defined in Eq. (95), the average energy and the entanglement can be obtained analytically. We emphasize that some of these quantities, such as the MSD, can be measured in the laboratory for ultracold gases [34]. Our first aim, is to solve the equation of motion (54). To obtain analytic expressions for $G_1(t)$, $G_2(t)$, we first consider the particular form of their Laplace transforms, Eqs. (55) and (56). They now read as

$$\mathcal{L}_z\left[\underline{\underline{G_1}}(t)\right] = \frac{z\mathbb{I} + \mathcal{L}_z\left[\underline{\underline{\Gamma}}(t)\right]}{z^2\mathbb{I} + z\mathcal{L}_z\left[\underline{\underline{\Gamma}}(t)\right]} = \frac{1}{z}\mathbb{I}, \tag{85a}$$

$$\mathcal{L}_z\left[\underline{\underline{G_2}}(t)\right] = \frac{1}{z^2\mathbb{I} + z\mathcal{L}_z\left[\underline{\underline{\Gamma}}(t)\right]}. \tag{85b}$$

Equation (85a) is independent of the specific form of the damping kernel, and can be easily inverted to get

$$G_1(t) = \mathbb{I}. \tag{86}$$

Equation (85b) depends on the form of the damping kernel. Thus, to get $G_2(t)$, one needs to compute the Laplace transform of the damping kernel, Eq. (63). In the long time limit, i.e. when $\mathrm{Re}[z] \ll \Lambda$, the Laplace transform of the damping kernel reads

$$\mathcal{L}_z\left[\Gamma_{jq}(t)\right] = z\tilde{\tau}_{jq}\Lambda + O\left(z^2\right), \tag{87}$$

and hence

$$\mathcal{L}_z\left[\underline{\underline{G_2}}(t)\right] = \frac{1}{\left(\mathbb{I} + \Lambda\underline{\underline{\tilde{\tau}}}\right)z^2}, \tag{88}$$

where

$$\underline{\underline{\tilde{\tau}}} = \left(\begin{array}{cc} \tilde{\tau}_{11} & \tilde{\tau}_{12} \\ \tilde{\tau}_{21} & \tilde{\tau}_{22} \end{array}\right).$$

This can be inverted as

$$\underline{\underline{G_2}}(t) = \frac{t}{\left(\mathbb{I} + \Lambda\underline{\underline{\tilde{\tau}}}\right)} = \frac{\left(\mathbb{I} + \Lambda\left(\begin{array}{cc} \tilde{\tau}_{22} & -\tilde{\tau}_{12} \\ -\tilde{\tau}_{21} & \tilde{\tau}_{11} \end{array}\right)\right)t}{(1 + \Lambda\tilde{\tau}_{11})(1 + \Lambda\tilde{\tau}_{22}) - \Lambda^2\tilde{\tau}_{12}\tilde{\tau}_{21}},$$

which we rewrite as

$$\underline{\underline{G_2}}(t) = \underline{\underline{\alpha}}t \quad \text{with} \quad \underline{\underline{\alpha}} = \left(\begin{array}{cc} \alpha_{11} & \alpha_{12} \\ \alpha_{21} & \alpha_{22} \end{array}\right), \tag{89}$$

with

$$\alpha_{jq} = \frac{\left(I_{jq} + (-1)^{j+q}\Lambda\tilde{\tau}_{jq}\right)}{(1 + \Lambda\tilde{\tau}_{11})(1 + \Lambda\tilde{\tau}_{22}) - \Lambda^2\tilde{\tau}_{12}\tilde{\tau}_{21}}. \tag{90}$$

Finally, the solutions of the equations of motion for the two impurities, written in the form of Eqs. (54), read as

$$x_j(t) = x_j(0) + \sum_{q=1}^{2}\alpha_{jq}\dot{x}_q(0)t + \int_0^t(t-s)\alpha_{jq}B_q(s)ds, \tag{91}$$

where

$$B_j(t) = \sum_{k\neq 0}i\hbar g_k^j\left[e^{i\omega_k t}b_k^\dagger(0) - e^{-i\omega_k t}b_k(0)\right].$$

Now, we can evaluate, first, the MSD for each one of the particles. The MSD is defined as

$$\left\langle\left[x_j(t) - x_j(0)\right]^2\right\rangle. \tag{92}$$

For the sake of simplicity, and to study the dynamical evolution of the MSD of the impurities purely due to their interaction with the bath, we assume that the initial states of the impurities and the bath are uncorrelated, $\rho(0) = \rho_I(0) \otimes \rho_B$, such that averages of the form $\left\langle\dot{x}_j(0)B_q(s)\right\rangle$, that would otherwise appear in the expression, vanish. In the results presented in Fig. 1, we assumed that there are no initial correlations between the two impurities such that the terms $\left\langle\dot{x}_j(0)x_q(0)\right\rangle$ and $\left\langle\dot{x}_j(0)\dot{x}_q(0)\right\rangle$ for $j \neq q$ also vanish. Nevertheless, the case of finite values for these expectation values, in particular all of them being equal to 1, was also considered without seeing a qualitative difference in the results. Furthermore, to evaluate the MSD one needs to evaluate

$$\left\langle\left\{B_j(t), B_q(s)\right\}\right\rangle = 2\hbar\nu_{jq}(t-s), \tag{93}$$

where

$$\nu_{jq}(t-s) = \Theta(t-s)\int_0^\infty J_{jq}(\omega)\coth\left(\frac{\hbar\omega}{2k_B T}\right)\cos(\omega(t-s))d\omega,$$

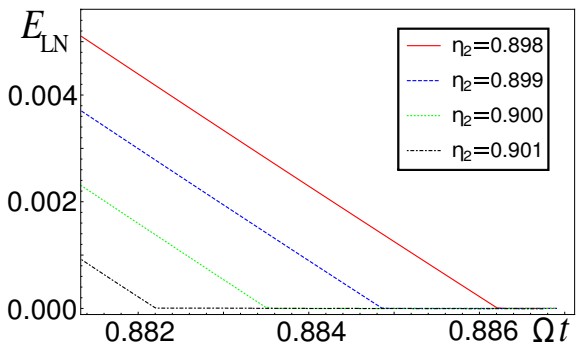

Figure 1: *Time dependence of the entanglement between the two kinds of untrapped impurities*. Entanglement, as evaluated using Eq. (75), is observed at the long but not infinite time limit, for the case of untrapped impurities of potassium K in a bath of particles of Rubidium Rb, at the low temperature limit. The initial variances of position and velocity for the two particles, as well as their covariances, are assumed to be 0, i.e. $\langle x_j^2(0) \rangle = \langle \dot{x}_j^2(0) \rangle = \langle x_j(0) x_q(0) \rangle = \langle \dot{x}_j(0) \dot{x}_q(0) \rangle = 0$ for $j \in \{1, 2\}$, however the qualitative behavior was the same for other initial conditions as well, namely for setting all quantities equal to a finite value (in particular equal to 1). Entanglement, decreases to zero as time passes. It is studied for a number of different coupling constants $\eta_2$ of the second impurity. The rest of the parameters are $\Omega = 2\pi \cdot 500Hz$, $\eta_1 = 1$, $g_B = 3 \cdot 10^{-37} J \cdot m$ and $n_0 = 7(\mu m)^{-1}$. It is observed that increasing $\eta_2$ decreases both the value of the entanglement and the time at which it reaches zero.

is the noise kernel. To prove this, we used the fact that for a bath mode at thermal equilibrium at a temperature $T$

$$\left\langle b_k^\dagger b_k \right\rangle = \frac{1}{e^{\frac{\hbar\omega}{k_B T}} - 1}.$$

Then the expression for the MSD of one of the particles, in the long time limit, takes the form

$$\left\langle \left[ x_j(t) - x_j(0) \right]^2 \right\rangle_{\rho(t)} = \alpha_{jj}^2 \left\langle \dot{x}_j^2(0) \right\rangle t^2$$
$$+ \frac{1}{2} \sum_{y,k=1}^{2} \frac{\alpha_{jk} \alpha_{jy}}{m_k m_y} \int_0^t ds \int_0^t d\sigma (t-s)(t-\sigma) \left\langle \left\{ B_k(t), B_y(s) \right\} \right\rangle. \tag{94}$$

In the regime of low temperatures, where $\coth\left( \frac{\hbar\omega}{2k_B T} \right) \approx 1$, the MSD becomes

$$\left\langle \left[ x_j(t) - x_j(0) \right]^2 \right\rangle_{\rho(t)} = \left( \alpha_{jj}^2 \left\langle \dot{x}_j^2(0) \right\rangle + \frac{1}{2} \sum_{y,k=1}^{2} \frac{\hbar \alpha_{jk} \alpha_{jy} \tilde{\tau}_{ky} \Lambda^2}{m_y m_k} \right) t^2. \tag{95}$$

Therefore, we find that the particles motion is superdiffusive. We note here that the same result was found for the single particle case [43], where this effect was attributed to the memory effects present in the system. In this context the result in Eq. (95) represents a witness of memory effects on a measurable quantity.

In [84, 85] the presence of memory effects is associated to backflow of energy. To examine whether such backflow of energy appears in our system as well, we derive an expression for the average energy of the system as a function of time. To do so, we need an expression for

the time evolution of the momentum, which reads as

$$p_j(t) = m_j \dot{x}_j(t) = m_j \left( \sum_{q=1}^{2} \alpha_{jq} \dot{x}_q(0) + \int_0^t \alpha_{jq} B_q(s) \, ds \right). \tag{96}$$

Thus, in the low temperature limit, the average energy as a function of time, reads as

$$E_j(t) = \frac{\langle p_j^2 \rangle}{2m_j} = \sum_{q,y=1}^{2} g_{\mathrm{IB}}^q n_0 + \alpha_{jq} E_q(0) + \frac{\hbar}{2} \alpha_{jq} \alpha_{jy} m_q \tilde{\tau}_{qy} \Lambda^2$$
$$- \hbar \alpha_{jq} \alpha_{jy} \tilde{\tau}_{qy} \frac{1}{t^2} [\cos(\Lambda t) + \Lambda t \sin(\Lambda t) - 1]. \tag{97}$$

The oscillatory behaviour of the energy suggests that, in addition to the traditional dissipation process where the impurity loses energy, also the environment provides energy to the impurity, *i.e.* we detect a backflow of energy from the environment to the impurity. We note that, in the two particle case, the diffusion coefficient in Eq. (95) is different for each particle [see expression for $\alpha_{jq}$, Eq. (90)]. Thus, it depends on the interactions of each kind of particle with the BEC and the mass of each particle, together with the density and coupling constant of the BEC [see expression for the cutoff frequency, $\Lambda$, Eq. (50)].

As explained in Sec. 2.4.1, we will use the logarithmic negativity to study entanglement and hence the covariance matrix of the two impurities is needed. To this end, we find for the low temperature case, $\coth(\hbar\omega/2k_{\mathrm{B}}T) \approx 1$ and in the long-time limit, $\mathrm{Re}[z] \ll \Lambda$

$$\langle x_j x_q \rangle = \sum_{k,y=1}^{2} \delta_{ky} \langle x_k(0) x_y(0) \rangle + \delta_{ky} \alpha_{jk} \alpha_{qy} \langle \dot{x}_k(0) \dot{x}_y(0) \rangle t^2 \tag{98a}$$
$$+ \frac{2\hbar m_j m_q \tilde{\tau}_{ky} \alpha_{jk} \alpha_{qy} (1 + \mathrm{mod}_2(i+j))}{m_k m_y} \left[ 2\gamma - 2 + \frac{(\Lambda t)^2}{2} + 2\cos(\Lambda t) - 2Ci(\Lambda t) + 2\log(\Lambda t) \right],$$

$$\langle x_j p_q \rangle = \tag{98b}$$
$$\sum_{k,y=1}^{2} \delta_{ky} m_q \alpha_{jk} \alpha_{qy} \langle \dot{x}_k(0) \dot{x}_y(0) \rangle t + \frac{\hbar m_j m_q \tilde{\tau}_{ky} \alpha_{jk} \alpha_{qy}}{m_y m_k t} \left( \frac{(\Lambda t)^2}{2} - [\cos(\Lambda t) + \Lambda t \sin(\Lambda t) - 1] \right),$$

$$\langle p_j p_q \rangle = \sum_{k,y=1}^{2} \delta_{ky} m_j m_q \alpha_{jk} \alpha_{qy} \langle \dot{x}_k(0) \dot{x}_y(0) \rangle \tag{98c}$$
$$+ 2m_j g_{\mathrm{IB}}^j n_0 \delta_{jq} + \frac{\hbar m_j m_q \tilde{\tau}_{ky} \alpha_{jk} \alpha_{qy}}{m_y m_k t^2} \left( \frac{(\Lambda t)^2}{2} - [\cos(\Lambda t) + \Lambda t \sin(\Lambda t) - 1] \right),$$

where in Eq. (98a), $Ci(x) = -\int_x^\infty \frac{\cos(t)}{t} dt$ is the cosine integral function.

In Fig. 1 we depict the entanglement as a function of time for the low temperature scenario and for different coupling constants $\eta_2$. We find entanglement in the long-time limit, which vanishes linearly. The maximum time at which entanglement reaches zero is increased by decreasing the interactions of the second kind of impurities. For a given time, entanglement increases with decreasing $\eta_2$. Also, for a given time we found that increasing $\eta_1$ while keeping the ratio $\eta_1/\eta_2$ constant, increases entanglement. We also studied the dependence of entanglement on density (not shown), finding that for large enough densities, the higher the density, the less entanglement was found and the faster it disappears. However, for low densities, increasing density increases entanglement. Since it is the presence of the bath that entangles

the particles, small density is necessary to induce entanglement. This is in accordance with the results for the trapped impurities, presented in next section. Quantitatively, entanglement of an order of magnitude larger was found for a density of an order of magnitude smaller than the one presented in Fig. 1. We note that, for each curve, there is a minimum time at which equipartition is fulfilled according to Eqs. (79). The form of $\nu_-$ can be analytically obtained from the expressions (98). We do not present it here explicitly to avoid long expressions.

Finally, the MSD for the relative motion $r = x_1 - x_2$ and center of mass $R = (x_1 + x_2)/2$ coordinates can also be obtained. We find that they equally perform superdiffusive motions. This means that both the variance of the distance between the two particles and the center of mass increases ballisticaly on average. For the particular case where the impurities are of the same mass and interact with the same strength with the BEC, the relative distance variance is constant in time. This is showing that it decouples from the bath. The center of mass position variance instead still grows ballisticaly with time. Instead, we find that for this case still one can find entanglement. This indicates that the vanishing of the entanglement at large times is not explained solely but the increase of the relative distance variance.

## 3.2 Squeezing and Entanglement for Trapped impurities

We conjecture that one should find squeezing and entanglement between the two particles in the regime where quantum effects play an important role. Thus, we consider the low temperature regime for the case of harmonically trapped particles, namely the regime where $k_B T \ll \hbar \Omega_j$ with $j = 1, 2$. The parameters that we use are such that the condition (39), that guarantees that no runaway solutions are encountered, is satisfied. At the same time the coupling constants used are relatively strong such that the non-Markovian effects are manifested. We find that for entanglement, one has to consider coupling constants in the range that satisfies:
$g_{\mathrm{IB}}^{(j)}/\Omega_j \in [0.01, 1]$, as below this range the bath effect is not enough to create entanglement, while above this, the effect of the bath destroys entanglement. In the trapped case we make use of the covariance matrix whose elements are constructed by numerical integration from Eqs. (66), (70) and (71). In general, and unless stated otherwise, we make the assumption that the centers of the harmonic traps are equal, $d_1 = d_2$, as entanglement is maximized in such case.

## 3.3 Squeezing

In this section we study squeezing as a function of the parameters of the system and the bath. To detect squeezing we make use of the condition (81). However, note that the value of $\nu_{min}$ is not a measure of the level of squeezing in the system, but a criterion that squeezing occurs. In the numerical computations presented in Fig. 2, we take $\Omega_1 = 600\pi Hz$, $\Omega_2 = 450\pi Hz$, $n_0 = 90(\mu m)^{-1}$, $\eta_1 = 0.325$ and $R_{12}/a_{\mathrm{HO}} = 0$, where distance was measured in units of a fixed harmonic oscillator length for both impurities equal to $\alpha_{\mathrm{HO}} = \sqrt{\hbar/(m\Omega)}$, with $m = m_1 = m_2$ and $\Omega = \pi kHz$ a typical frequency of the same order of magnitude as the frequencies considered throughout all of our studies. Without loss of generality, we assumed $d_1 \geq d_2$ such that $R_{12} \geq 0$. In our studies, we varied the temperature $T$, $\eta_2$ and $g_B$. The qualitative behaviour for a varying $n_0$ was found to analogous to that of $g_B$, so the results with respect to this variable are not shown. The parameters used are within current experimental feasibility [34]. In Fig. 2(a) we studied squeezing as a function of temperature for a number of different $\eta_2$ and, in panel (b), squeezing as a function of the temperature for various $g_B$. First, we find squeezing at low temperatures, in the $nK$ regime, that vanishes at higher temperatures. Second, we observe that the temperature at which squeezing vanishes increases with the coupling constant $\eta_2$ or $g_B$. Furthermore, as we will show in next section, squeezing appears in the range of temper-

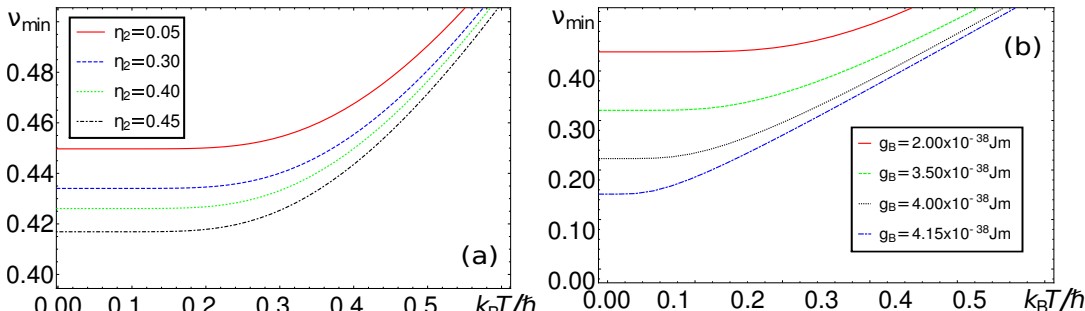

Figure 2: *Temperature dependence of the squeezing between the two kinds of trapped impurities*. In (a) we study the temperature dependence of squeezing for different values of $\eta_2$ with $g_B = 2.36 \cdot 10^{-38} J \cdot m$ and in (b) for different values of $g_B$ with $\eta_2 = 0.295$. In both cases we use impurities of potassium K in a bath of particles of Rubidium Rb and we set: $\Omega_1 = 600\pi Hz$, $\Omega_2 = 450\pi Hz$, $n_0 = 90(\mu m)^{-1}$, $\eta_1 = 0.325$ and $R_{12}/a_{HO} = 0$.

atures where entanglement appears as well, as can be seen in Fig. 3, but the range is slightly larger than that for entanglement. The existence of a relation of squeezing and entanglement is in agreement for example with the work in [86], where an ohmic spectral density was considered, such that analytic results could be obtained, and they find in the long-time limit that the logarithmic negativity was given by $\mathcal{E}_{LN}(\rho_{12}) = 2r$ with $r$ being the two-mode squeezed state squeezing parameter.

We also studied the position and momentum variances,

$$\delta_{x_j} = \sqrt{\frac{2m_j\Omega_j\left\langle x_j^2\right\rangle}{\hbar}}, \quad \delta_{p_j} = \sqrt{\frac{2\left\langle p_j^2\right\rangle}{m_j\Omega_j\hbar}}, \tag{99}$$

observing that indeed in the large temperature limit they approach the equipartition theorem. This means that, in this limit, the system is formally analogous to two independent harmonic oscillators as expected. We used this as a test to verify the validity of our numerical results. In appendix D we study the equilibrium Hamiltonian in detail. This allows us to find a prediction for the large temperature limit of, e.g., $\langle x_1 x_2 \rangle$ or $\langle p_1 p_2 \rangle$. We found that these correlation functions do not vanish at large $T$, not implying the presence of quantum correlations but only classical correlations in this limit. This is in agreement with the fact that one only finds entanglement for very low $T$.

In addition, we calculated the uncertainty $\delta_{x_1}$ when compared to $\delta_{p_1}$, restricting only to the regime where Heisenberg uncertainty principle was fulfilled. This amounts to studying squeezing for the partially traced state of the system, tracing out the other kind of impurities. This way we were able study how the introduction of a second kind of impurities modified the squeezing found when defined as that of only one impurity (as is done in [43]). We found that the squeezing observed for one particle reduces as $\eta_2$ increases.

## 3.4 Thermal entanglement induced by isotropic substrates

Here we study the appearance of thermal entanglement, i.e. assuming that the entangled resource, the bath, connecting the two impurities is in a canonical Gibbs ensemble density matrix at certain $T$. In general the following parameters were used: $\Omega_1 = 600\pi Hz$, $\Omega_2 = 450\pi Hz$, $\eta_1 = 0.325$, $\eta_2 = 0.295$, $n_0 = 90(\mu m)^{-1}$, $T = 4.35 nK$, $g_B = 2.36 \cdot 10^{-38} J \cdot m$ and $R_{12}/a_{HO} = 0$.

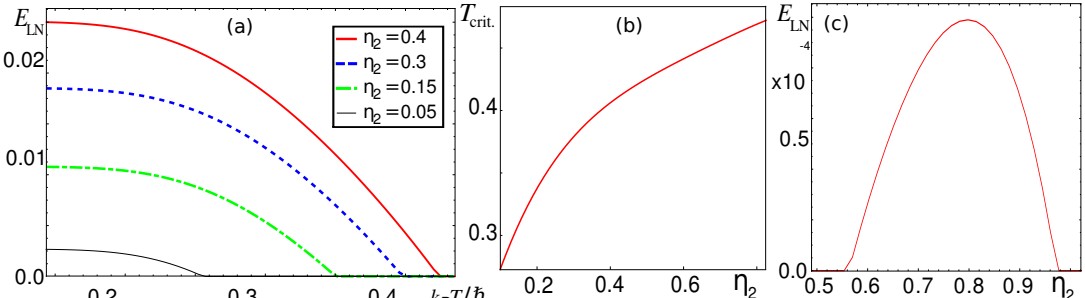

Figure 3: *Coupling strength dependence of the entanglement between the two kinds of trapped impurities*. (a) Entanglement as a function of $T$ for various couplings. As shown, for the values of $\eta_2$ considered here, increasing the interactions of the second type of impurities, $\eta_2$, enhances entanglement, at fixed $T$. For increasing $T$, entanglement decreases and eventually vanishes at certain $T_{\text{crit}}$. (b) The $T_{\text{crit}}$ at which the entanglement goes to zero increases with $\eta_2$ and seems to saturate at large $\eta_2$. (c) the entanglement at fixed $T = 4.35 nK$ increases for a range of $\eta_2$ and then decreases to zero. In this case we considered $n_0 = 350(\mu m)^{-1}$ and $g_B = 9.75 \cdot 10^{-39} J \cdot m$. In all of the graphs, we are considering impurities of potassium K in a bath of particles of Rubidium Rb. The parameters used in these plots are: $\Omega_1 = 600\pi Hz$, $\Omega_2 = 450\pi Hz$, $\eta_1 = 0.325$, $n_0 = 90(\mu m)^{-1}$, $g_B = 2.36 \cdot 10^{-38} J \cdot m$ and $R_{12}/a_{\text{HO}} = 0$

In certain cases we consider other values of $n_0$ and $g_B$ to study the appearance of the phenomenon of the bath causing a decrease of the entanglement. Also, some general comments about the results that will be presented below, are the following. First, it was observed that parameter regimes existed in which the uncertainty principle, translated into the condition Eq. (79), was not satisfied. We only present results when this condition is fulfilled. In the results presented here, entanglement is normalized based on the instance of maximum entanglement found in the system, which was obtained for the following parameters: $\Omega_1 = 600\pi Hz$, $\Omega_2 = 450\pi Hz$, $\eta_1 = 0.325$, $\eta_2 = 0.295$, $n_0 = 90(\mu m)^{-1}$, $T = 0.0435 nK$, $g_B = 4.25 \cdot 10^{-38} J \cdot m$ and $R_{12}/a_{\text{HO}} = 0$, and the maximum value of entanglement obtained was $\mathcal{E}_{\text{LN}} = 1.025$.

For all figures where we show the dependence of entanglement on temperature, we emphasize that below a certain temperature, the uncertainty principle was not satisfied. This minimum temperature depends on the other parameters of the system. For example, the larger the distance considered was, the lower temperatures that one can reach under the requirement that the uncertainty principle holds. We also note that for low enough temperatures, we find numerically a saturation of entanglement in all cases. Finally, the general behaviour of entanglement with temperature is to decrease, as expected, and beyond a certain temperature it vanishes. We term this as critical temperature, $T_{\text{crit}}$

In Fig. 3 (a), the temperature dependence of entanglement was studied for a number of coupling constants $\eta_2$, and it was observed that increasing $\eta_2$ implied an increase in entanglement, as well as an increase in $T_{\text{crit}}$. Information about this temperature is particularly important experimentally, and for this reason we studied the dependence of $T_{\text{crit}}$ on $\eta_2$ in Fig. 3 (b). We see that the increase of $T_{\text{crit}}$ with $\eta_2$ is decreasing for larger $\eta_2$. The dependence of entanglement was studied also as a function of $\eta_2$. In this case we considered $n_0 = 350(\mu m)^{-1}$ and $g_B = 9.75 \cdot 10^{-39} J \cdot m$ which allowed us to see the diminishing effect of the bath, meaning that entanglement reached a peak value for some $\eta_2$ and was later then decreases with increasing values of $\eta_2$.

In Fig. 4 (a), we study entanglement as a function of the temperature for various distances between the two impurities. As expected, entanglement decreases with increasing distance.

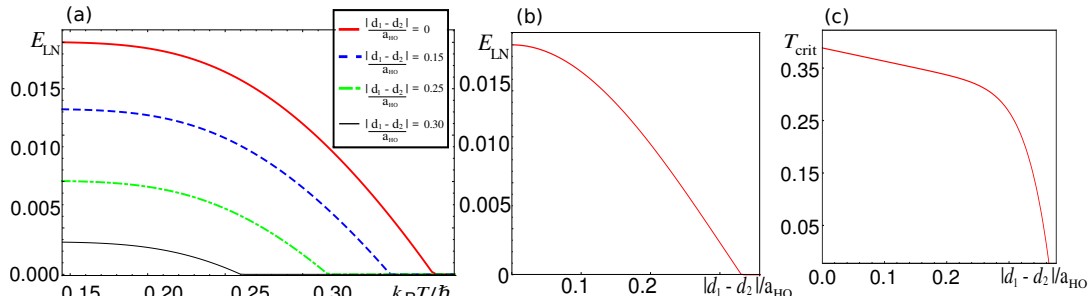

Figure 4: *Distance dependence of the entanglement between the two kinds of impurities.* In (a) we study entanglement as a function of $T$ for various distances between the trap of each kind of impurity. As expected the entanglement is reduced as the distance is increased. This is shown in figure (b). For each distance, at certain $T$ the entanglement vanishes. The dependence of this critical temperature as a function of distance is shown in (c). In all of the graphs, we are considering impurities of potassium K in a bath of particles of Rubidium Rb. The parameters used in these plots are: $\Omega_1 = 600\pi Hz$, $\Omega_2 = 450\pi Hz$, $\eta_1 = 0.325$, $\eta_2 = 0.295$, $n_0 = 90(\mu m)^{-1}$, $T = 4.35nK$ and $g_B = 2.36 \cdot 10^{-38} J \cdot m$.

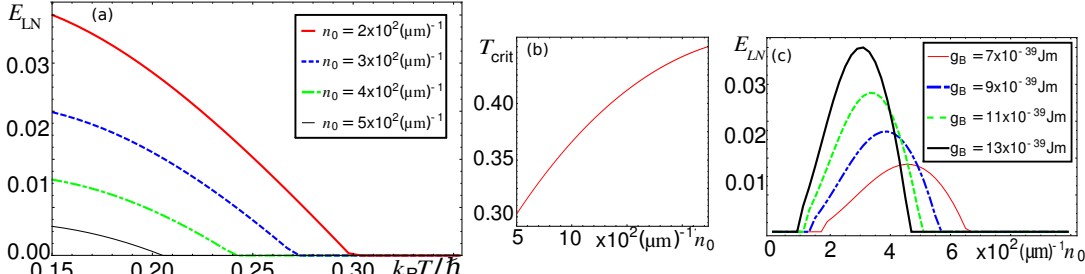

Figure 5: *Dependence of the entanglement between the two kinds of impurities on the density of the bosons in the bath.* In (a) we study entanglement as a function of $T$ for various densities of bosons. As shown, in this range of densities, increasing the density of bosons enhances entanglement. For each density, at certain $T_{crit}$, the entanglement vanishes. The dependence of $T_{crit}$ as a function of density is shown in (b). In (c) we illustrate that the entanglement at fixed $T$ increases for a range of $n_0$ and then decreases to zero, as it was the case with $\eta_2$ in Fig. 1. We show this dependence for various values of the coupling constant among the bosons. As shown, increasing $g_B$ increases the maximum value of the entanglement reached, but also entanglement vanishes at a smaller value of the density. In all of the graphs, we are considering impurities of potassium K in a bath of particles of Rubidium Rb. The parameters are: $\Omega_1 = 600\pi Hz$, $\Omega_2 = 450\pi Hz$, $\eta_1 = 0.325$, $\eta_2 = 0.295$, $T = 4.35nK$, $g_B = 2.36 \cdot 10^{-38} J \cdot m$ and $R_{12}/a_{HO} = 0$.

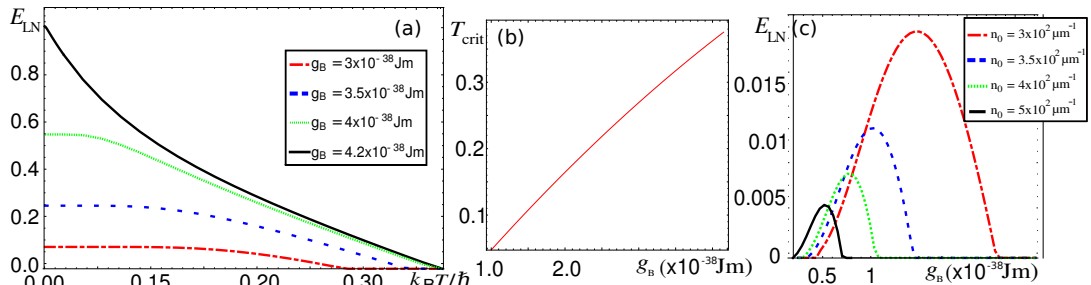

Figure 6: *Dependence of the entanglement between the two kinds of impurities on the coupling constant between the bosons $g_B$. In (a) we study entanglement as a function of $T$ for various $g_B$. As can be seen for the largest considered value of $g_B$, namely $g_B = 4.2 \cdot 10^{-38} J \cdot m$, the qualitative behaviour of entanglement with temperature, changes. For each density, at certain $T_{crit}$, the entanglement vanishes. The dependence of $T_{crit}$ as a function of density is shown in (b). In (c) we illustrate that the entanglement at fixed $T$ increases for a range of $g_B$ and then decreases to zero, as it was the case with $\eta_2$ in Fig. 1. We show this dependence for various values of the density of the bosons. As shown, in this range of parameters, decreasing $n_0$ increases the maximum value of the entanglement reached, but also entanglement vanishes at larger values of $g_B$. In all of the graphs, we are considering impurities of potassium K in a bath of particles of Rubidium Rb. The parameters used in these plots are: $\Omega_1 = 600\pi Hz$, $\Omega_2 = 450\pi Hz$, $\eta_1 = 0.325$, $\eta_2 = 0.295$, $n_0 = 90(\mu m)^{-1}$, $T = 4.35 nK$ and $R_{12}/a_{HO} = 0$.*

Furthermore, in Fig. 4 (c) the $T_{crit}$ also decreases with distance and it acquires a maximum for distance equal to 0. In Fig. 4 (b) we see that entanglement drops to 0 beyond a certain distance which is $R_{12} \sim 0.35 a_{HO}$, which for the parameters that we have chosen results in $0.2\mu m$. The distance at which entanglement drops to 0 depends on the other parameters of the system as well, e.g. it increases with decreasing temperature, but remains at the same order of magnitude.

In Fig. 5 (a), we show the dependence of entanglement with $T$ for various densities. The dependence of $T_{crit}$ on the density in Fig. 5 (b) again shows that it increases with $n_0$ for small densities. However, in Fig. 5 (c) we show that, while for small densities entanglement grows with the density, for larger values of the density it starts to decrease toward zero. We plot this figure for various values of $g_B$, showing that the larger the bosons interactions the smaller the value of $n_0$ at which the entanglement starts to decrease to zero.

Similar studies with similar results were undertaken for the dependence of the system on $g_B$ and are presented in Fig. 6. There, for $g_B = 4.2 \cdot 10^{-38} J \cdot m$ we see the qualitative behaviour of entanglement with a varying temperature. These results, together with the fact that we do not assume a particular form for the state of the two impurities initially, are clear indications that the induced entanglement between the two impurities is an effect of their interaction with the common bath.

In Fig. 7 (a) we present the results of the dependence of entanglement with $T$ for various ratios $\Omega_2/\Omega_1$. In Fig. 7 (b) we observe that $T_{crit}$ is not monotonically dependent on the ratio of trapping frequencies. This implies that there is a regime where, even though one keeps the temperature constant at a value where there is no entanglement, by increasing the trapping potential, such that the second impurity is more confined, entanglement appears for this given temperature. In Fig. 7 (c) we see that at resonance ($\Omega_2/\Omega_1 = 1$), entanglement achieves its maximum. This was also observed in Ref. [57]. The reason is that, since the

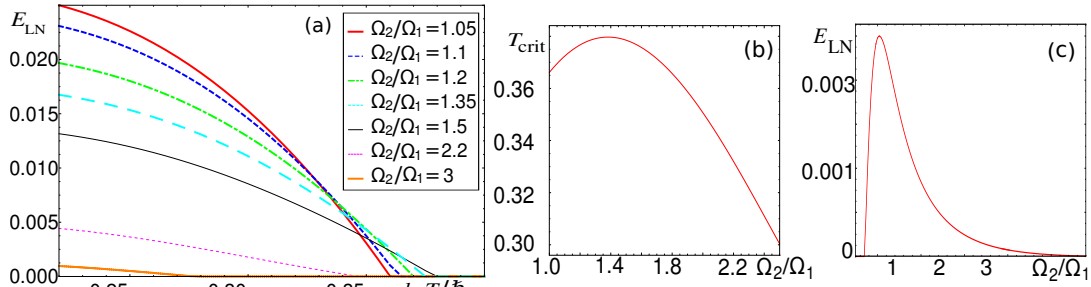

Figure 7: *Trapping frequency dependence of the entanglement between the two kinds of impurities.* In (a) we study entanglement as a function of $T$ for various ratios between the trap frequencies of each kind of impurity. For each $\Omega_2/\Omega_1$, at certain $T_{\text{crit}}$, the entanglement vanishes. The dependence of $T_{\text{crit}}$ as function of $\Omega_2/\Omega_1$ is shown in (b). (c) illustrates that the entanglement at fixed $T$ has a maximum value for certain value of $\Omega_2/\Omega_1$. The parameters $n_0 = 500(\mu m)^{-1}$ and $g_B = 9.75 \cdot 10^{-39} J \cdot m$ were used in this case. In all of the graphs, we are considering impurities of potassium K in a bath of particles of Rubidium Rb. The parameters used in these plots are: $\Omega_1 = 600\pi Hz$, $\eta_1 = 0.325$, $\eta_2 = 0.295$, $n_0 = 90(\mu m)^{-1}$, $T = 4.35 nK$, $g_B = 2.36 \cdot 10^{-38} J \cdot m$ and $R_{12}/a_{\text{HO}} = 0$.

values of the entries of the matrix $\underline{\Gamma}(t)$ we use are so small, what guarantees that the equations of motion for the two harmonic oscillators cannot be decoupled into two equations of motion for the center of mass and relative distance degrees of freedom, is the fact that the two trapping frequencies are different. To clarify this point, and draw the parallel with the results in [57], we note that at relatively high temperatures, in which however entanglement can still be observed, one could assume that the system is described by the effective Hamiltonian Eq. (110) in Appendix D, where the effect of the bath degrees of freedom is represented by an effective coupling between the two particles. In the limit $K/\Omega_j \to 0$ with $j \in 1, 2$, for $\Omega_1 = \Omega_2$, the aforementioned decoupling takes place, and the state of the system goes to a non-symmetric two-mode squeezed thermal state with infinite squeezing, i.e. the ideal EPR (maximally entangled) state. We finally note that, as discussed in Appendix D, it is possible to find an approximate prediction for the critical $T$, given by Eq. (112). We found that this prediction is in the same order of magnitude (nK) for all numerical results presented.

## 4 Conclusions

In this paper, we studied the emergent entanglement between two distinguishable polarons due to their common coupling to a BEC bath. To this end, we formulated the problem of two different kinds of impurities immersed in a BEC as a quantum Brownian motion model. The BEC is assumed to be confined in one dimension and homogeneous. The impurities do not interact among themselves, but only with the BEC. By means of a Bogoliubov transformation we diagonalize the part describing the BEC in the Hamiltonian. This brings the total Hamiltonian into a form in which one can identify the BEC part as a collection of oscillators with different frequencies, thus resembling a bath Hamiltonian. Also, we identify the impurities part of the Hamiltonian as the system Hamiltonian, in the usual terminology from open quantum systems. Finally, under the same physical constraints discussed in [43] for the Bose polaron problem, we linearise the interaction Hamiltonian, which brings the Hamiltonian into a conventional Caldeira-Leggett-like Hamiltonian, i.e., a quantum Brownian motion model.

We henceforth solve the associated coupled quantum Langevin equations system of motion, which encode the bath as a damping and a noise kernel. The damping kernel includes a non-diagonal term, often called hydrodynamic term in the context of Brownian particles, as it encodes the effect of the particles on one another. We find that the spectral density characterizing the bath is superohmic in 1D. We emphasize that in our work the properties of the bath, and particularly the spectral density, are not arbitrarily assumed but derived from physical considerations of the BEC. We solve these equations both for the case of untrapped and trapped impurities. We do not add artificial terms to the Hamiltonian as to make it non-negative. Instead we find a condition on the parameters of the system, for which the energy spectrum of the Hamiltonian is positive.

For the untrapped case we were able to solve the equations of motion analytically. Hence, we studied the MSD and the diffusive properties of the impurities, which are found to perform a superdiffusive motion. In addition, we studied the momentum variance of the impurities, observing an energy back-flow from the bath, attributed to its non-Markovian nature. Moreover, we obtained the covariance matrix explicitly. From the covariance matrix, we quantified entanglement between the two types of impurities using the logarithmic negativity. This is found to decrease linearly as a function of time. What is more, the relative distance and center of mass coordinates were considered for the case of identical impurities. The former becomes decoupled from the bath, such that it no more performs a superdiffusive motion, hence the variance of the distance between the impurities stays on average constant. Yet, we can detect a linearly decreasing with time entanglement between the two types of impurities, so we conjecture that the decrease of entanglement is attributed to their interaction with the bath rather than them running away from each other. This conjecture is further enhanced from our studies of the entanglement dependence on the rest of the parameters of the system, i.e. the density of the bosons $n_0$ in the BEC and their interaction strength $g_B$. In particular, we found that for any fixed finite time, and fixed $n_0$ ($g_0$), entanglement reaches a maximum value at some optimal $g_0$ ($n_0$). Increasing this value beyond the optimal reduces the entanglement until it vanishes.

For the trapped case, we obtained the covariance matrix elements numerically. We saw that the coherence correlations (off-diagonal terms of the covariance matrix) were linearly increasing with temperature, unless the parameters of the impurities were the same. Nevertheless, entanglement decreases as a function of temperature, which means that these correlations are not quantum. In the case of trapped impurities, entanglement was studied in detail as a function of the rest of the parameters of the system. It was found to decrease with increasing distance between the centers of the two trapping potentials. Furthermore, entanglement was found to increase and then decrease as a function of both $n_0$ and $g_B$. Moreover, entanglement is maximized at resonance of the frequencies of the two trapping potentials as was seen in previous similar studies as well [57].

Beyond entanglement, for all of these parameters the dependence of the critical temperature was also studied. In Appendix D a rough estimate of the critical temperature is made, using the effective form of the Hamiltonian in the thermalized regime. The estimate is of the same order of magnitude as the critical temperature observed in our studies. Squeezing was also examined in this case as a function of all the parameters of the system, found to behave qualitatively the same as entanglement. In particular, it is seen that entanglement always appears if there is squeezing but the converse is not always true.

In summary, we have studied the emergence of entanglement of two types of impurities embedded in the same bath, starting from a Hamiltonian justified on physical grounds. We examined analytically the case of two untrapped impurities, and gave numerical results in the scenario of harmonically trapped impurities. The dependence of entanglement, squeezing as well as the critical temperature, i.e. the temperature beyond which entanglement vanishes,

were studied as functions of the physical parameters of the system. The parameters of the system used were within current experimental settings and we believe that our results can be experimentally verified. These results on squeezing and entanglement in these setups are particularly interesting as the two phenomena represent resources for quantum information processing.

## Acknowledgements

Insightful discussion with A.A. Valido and S. Kohler are gratefully acknowledged.

**Funding information**     This work has been funded by a scholarship from the Programa Màsters d'Excel·lència of the Fundació Catalunya-La Pedrera, the Spanish Ministry MINECO (National Plan 15 Grant: FISICATEAMO No. FIS2016-79508-P, SEVERO OCHOA No. SEV-2015-0522), Fundació Privada Cellex, Generalitat de Catalunya (AGAUR Grant No. 2017 SGR 1341 and CERCA/Program), ERC AdG OSYRIS, EU FETPRO QUIC, and the National Science Centre, Poland-Symfonia Grant No. 2016/20/W/ST4/00314. M.M. acknowledges financial support from the Spanish Ministry MINECO (project QIBEQI FIS2016-80773-P).

## A  Spectral density

In this appendix we briefly present the derivation of the spectral density for a continuous spectrum of Bogoliubov modes, given in Eq. (51), following the work in [43]. In particular, the sum in Eq. (42) is turned into the integral

$$\sum_{k \neq 0} \rightarrow \int \frac{V}{(2\pi)^d} d^d k,$$

and using the relation

$$\delta(\omega - \omega_k) = \frac{1}{\partial_{\mathbf{k}} \omega_{\mathbf{k}}\,|_{\mathbf{k}=\mathbf{k}_\omega}} \delta(\mathbf{k} - \mathbf{k}_\omega),$$

we obtain

$$J_{jq}(\omega) = \frac{n_0 g_{\mathrm{IB}}^{(j)} g_{\mathrm{IB}}^{(q)} S_d}{\hbar (2\pi)^d} \int d\mathbf{k}\, \mathbf{k}^{d+1} \sqrt{\frac{(\xi \mathbf{k})^2}{(\xi \mathbf{k})^2 + 2}} \frac{\cos(\mathbf{k} R_{12})}{\partial_{\mathbf{k}} \omega_{\mathbf{k}}\,|_{\mathbf{k}=\mathbf{k}_\omega}} \delta(\mathbf{k} - \mathbf{k}_\omega).$$

For a 1D environment $S_1 = 2$, so that the continuous form of the spectral density (51) is obtained.

## B  Susceptibility

Here we evaluate the form of the susceptibility for the spectral density, given in Eq. (53). The imaginary part of the susceptibility is simply given by

$$Im[\lambda_{jq}(\omega)] = -\hbar(\Theta(\omega) - \Theta(-\omega)) J_{jq}(\omega),$$

while the real part is given by

$$
\begin{aligned}
Re\big[\lambda_{jq}(\omega')\big] &= \mathcal{H}\big[Im\big[\lambda_{jq}(\omega)\big]\big](\omega') \\
&= \frac{1}{\pi}P\int_{-\infty}^{\infty}\frac{Im\big[\lambda_{jq}(\omega)\big]}{\omega-\omega'}d\omega \\
&= -\frac{\hbar}{\pi}P\int_{-\infty}^{\infty}\frac{(\Theta(\omega)-\Theta(-\omega))J_{jq}(\omega)}{\omega-\omega'}d\omega \\
&= -\frac{\hbar\widetilde{\tau}_{jq}}{\pi}P\int_{-\infty}^{\infty}\frac{(\Theta(\omega)-\Theta(-\omega))\,\omega^3\cos\left(\frac{\omega}{c}R_{jq}\right)e^{-\frac{\omega}{\Lambda}}}{\omega-\omega'}d\omega \\
&= -\frac{\hbar\widetilde{\tau}_{jq}}{\pi}P\int_{0}^{\infty}\omega^3\cos\left(\frac{\omega}{c}R_{jq}\right)\left(\frac{1}{\omega-\omega'}+\frac{1}{\omega+\omega'}\right)d\omega,
\end{aligned}
$$

where $\mathcal{H}$ represents the Hilbert transform, $\Theta$ is the heaviside step function and $P$ is the principal number. We first find

$$
P\int_{0}^{\infty}\omega^3\frac{e^{-\frac{\omega}{\Lambda}}\cos\left(\frac{\omega}{c}R_{jq}\right)}{\omega-\omega'}d\omega,
$$

as we can evaluate it with the property of the Hilbert transform

$$
\mathcal{H}[\omega f(\omega)] = \omega\mathcal{H}[f(\omega)] + \frac{1}{\pi}\int_{-\infty}^{\infty}f(\omega)d\omega.
$$

We apply this property three times to obtain

$$
P\int_{0}^{\infty}\omega^3\frac{e^{-\frac{\omega}{\Lambda}}\cos\left(\frac{\omega}{c}R_{jq}\right)}{\omega-\omega'}d\omega = \omega'^3 P\int_{0}^{\infty}\frac{e^{-\frac{\omega}{\Lambda}}\cos\left(\frac{\omega}{c}R_{jq}\right)}{\omega-\omega'}d\omega + \omega'^2\frac{\Lambda}{1+\left(\Lambda\frac{R_{jq}}{c}\right)^2}
$$

$$
+ \omega'\frac{\frac{1}{\Lambda^2}-\frac{R_{jq}^2}{c^2}}{\left(\frac{1}{\Lambda^2}+\frac{R_{jq}^2}{c^2}\right)^2} + 2\frac{\left(\frac{1}{\Lambda^3}-3\frac{R_{jq}^2}{\Lambda c^2}\right)}{\left(\frac{1}{\Lambda^2}+\frac{R_{jq}^2}{c^2}\right)^3}. \tag{100}
$$

We can then also show that:

$$
P\int_{0}^{\infty}\frac{e^{-\frac{\omega}{\Lambda}+i\frac{\omega}{c}R_{jq}}}{\omega-\omega'}d\omega = \begin{cases} e^{-\frac{\omega'}{\Lambda}+i\frac{\omega'}{c}R_{jq}}\left(\Gamma\left[0,-\frac{\omega'}{\Lambda}+i\frac{\omega'}{c}R_{jq}\right]+i\pi\right) \\ e^{-\frac{\omega'}{\Lambda}+i\frac{\omega'}{c}R_{jq}}\Gamma\left[0,-\frac{\omega'}{\Lambda}+i\frac{\omega'}{c}R_{jq}\right] \end{cases},
$$

where the top case corresponds to $\omega' \in (0,\infty)$ and the bottom to the complementary interval $\omega' \in (-\infty,0)$. Here $\Gamma[\alpha,z] = \int_{z}^{\infty}t^{\alpha-1}e^{-t}dt$ denotes the upper incomplete gamma function. After introducing (100) in the expression for the real part of the susceptibility above, we obtain that

$$
Re\big[\lambda_{jq}(\omega')\big] = -\frac{\hbar\widetilde{\tau}_{jq}}{\pi}\Bigg\{\omega'^3 Re\big[g(\omega')-g(-\omega')\big]
$$

$$
+ \pi\omega'^3 Im\left[\Theta(\omega')e^{-\frac{\omega'}{\Lambda}+i\frac{\omega'}{c}R_{jq}} + (\omega'\to-\omega')\right] + 2\omega'^2\frac{\Lambda}{1+\left(\Lambda\frac{R_{jq}}{c}\right)^2} + 4\frac{\left(\frac{1}{\Lambda^3}-3\frac{R_{jq}^2}{\Lambda c^2}\right)}{\left(\frac{1}{\Lambda^2}+\frac{R_{jq}^2}{c^2}\right)^3}\Bigg\}, \tag{101}
$$

where $(\omega' \to -\omega')$ stands for $\Theta(-\omega') e^{\frac{\omega'}{\Lambda} - i\frac{\omega'}{c}R_{jq}}$ and

$$g(\omega') = e^{-\frac{\omega'}{\Lambda} + i\frac{\omega'}{c}R_{jq}}\Gamma\left[0, -\frac{\omega'}{\Lambda} + i\frac{\omega'}{c}R_{jq}\right].$$

With this, the susceptibility takes the form in Eq. (73).

## C   Study of an exact expression for the covariance matrix elements

In the particular case of $R_{12} = 0$, it can be shown that the integral in Eq. (59) takes the following form:

$$\int_{-\infty}^{+\infty} \frac{g_6(\omega)}{h_6(\omega)h_6(-\omega)} d\omega, \tag{102}$$

where $h_6(\omega)$ is a 6th order polynomial, and $g_6(\omega)$ is a 7th order polynomial. An integral of this form was also obtained in [57], for an ohmic spectral density. The criterion to use this formula, is that the roots of $h_6(\omega)$ lie in the upper half plane. Finding the roots of this polynomial requires finding expressions for 12 variables, 6 real and 6 imaginary. In our case one can show that the polynomial $h_6(\omega)$ can be written as

$$h_6(\omega) = \sum_{j=0}^{6} A_j (2\pi i\omega)^{6-j}, \tag{103}$$

with the coefficients $A_j$ being all real, which implies that the roots of $h_6(\omega)$ are symmetrically located about the imaginary axis. This reduces the problem in finding just 6 variables, 3 real and 3 imaginary. Furthermore, by making use of the Vieta relations, one can show that

$$\frac{Im[z_1]}{|z_1|^2} + \frac{Im[z_2]}{|z_2|^2} + \frac{Im[z_3]}{|z_3|^2} = 0. \tag{104}$$

This implies that not all roots of the polynomial can lie in the upper half plane. The reason for this can be traced back to the fact that there is no linear term in the polynomial $h_6(\omega)$, which is an artefact of considering a super-ohmic spectral density as can be understood by comparing to the case in [57]. The issue of not all roots being in the upper half plane, can be resolved in the following way. The roots of $h_6(\omega)$ that are in the lower half plane, have mirrored roots in the upper half plane in the polynomial $h_6(-\omega)$. If the expressions for the roots could be derived, one could simply redefine polynomials $h_6(\omega)$ and $h_6(-\omega)$ to $\tilde{h}_6(\omega)$ and $\tilde{h}_6(-\omega)$ such that all the roots of $\tilde{h}_6(\omega)$ would lie in the upper half plane and all the roots of $\tilde{h}_6(-\omega)$ would lie in the lower half plane. However for a 6th order polynomial of the form we have, one cannot find the roots. So this could perhaps only be applied in an algorithmic way, where one first selects a set of parameters and then makes the redefinition of the polynomials once the roots are found.

## D   Equilibrium Hamiltonian

Here we discuss the thermalization properties of the system. We will use the fact that the two kinds of impurities are formally analogous to two oscillators. The bath dependent part of the

Hamiltonian (18) is

$$\tilde{U}\left(x_1, x_2, \{b_k^\dagger, b_k\}_{k\neq 0}\right) = i \sum_{\substack{j=1 \\ k\neq 0}}^{2} \hbar g_k^{(j)}\left(e^{ikd_j}b_k - e^{-ikd_j}b_k^\dagger\right)x_j + \sum_{k\neq 0} E_k b_k^\dagger b_k.$$

For the system to thermalize, there should be no memory effects induced on the oscillator due to its coupling with the thermal bath. This means that the Hamiltonian will have to be independent of the bath variables $\{b_k^\dagger, b_k\}_{k\neq 0}$ evolution, i.e. $\forall k \neq 0$ the following should be fulfilled

$$\frac{\partial \tilde{U}\left(x_1, x_2, \{b_k^\dagger, b_k\}_{k\neq 0}\right)}{\partial b_k} = \frac{\partial \tilde{U}\left(x_1, x_2, \{b_k^\dagger, b_k\}_{k\neq 0}\right)}{\partial b_k^\dagger} = 0.$$

This results in the following conditions

$$b_k(t) = \frac{i\hbar}{E_k} \sum_{j=1}^{2} g_k^{(j)} e^{-ikd_j} x_j(t), \tag{105}$$

$$b_k^\dagger(t) = -\frac{i\hbar}{E_k} \sum_{j=1}^{2} g_k^{(j)} e^{ikd_j} x_j(t). \tag{106}$$

Replacing these expressions, for the bath degrees of freedom operators, in the initial Hamiltonian (18), it becomes

$$H_{\text{Lin}} = \sum_{j=1}^{2}\left[\frac{p_j^2}{2m_j} + \frac{1}{2}m_j\Omega_j^2\left(x_j + d_j\right)^2\right] + W(d_1, d_2)(x_1 + x_2)$$

$$- 2\hbar^2 \sum_{\substack{q,j=1 \\ k\neq 0}}^{2} \frac{1}{E_k} g_k^{(q)} g_k^{(j)} \cos\left(d_j - d_q\right) x_j x_q.$$

This can be rewritten as

$$H_{\text{Lin}} = \sum_{j=1}^{2}\left[\frac{p_j^2}{2m_j} + \frac{1}{2}m_j\left(\Omega_j^2 - \tilde{\Omega}_j^2\right)x_j^2\right] - 2Kx_1x_2 + \widehat{W}(d_1, d_2, x_1, x_2), \tag{107}$$

where $\tilde{\Omega}_j^2 = \frac{1}{m_j} 2\hbar^2 \sum_{k\neq 0} \frac{1}{E_k}\left(g_k^{(j)}\right)^2$, $K = 2\hbar^2 \sum_{k\neq 0} \frac{1}{E_k} g_k^{(1)} g_k^{(2)} \cos(d_1 - d_2)$ and $\widehat{W}(d_1, d_2, x_1, x_2)$ is the function of the remaining constant terms and terms linear in $x_1$ and $x_2$. At thermal equilibrium, the terms in $\widehat{W}(d_1, d_2, x_1, x_2)$ will not affect the equilibrium state. Then, one can neglect them and consider the effective Hamiltonian

$$H_{\text{eff}} = \sum_{j=1}^{2}\left[\frac{p_j^2}{2m_j} + \frac{1}{2}m_j\left(\Omega_j^2 - \tilde{\Omega}_j^2\right)x_j^2\right] - 2Kx_1x_2. \tag{108}$$

This is formally analogous to two effectively coupled harmonic oscillators. To decouple them, we transform to the normal modes of the system, $Q_1, Q_2$. This is achieved by an orthogonal transformation of the form

$$\begin{pmatrix} Q_1 \\ Q_2 \end{pmatrix} = \begin{pmatrix} \cos(\theta) & -\sin(\theta) \\ \sin(\theta) & \cos(\theta) \end{pmatrix}\begin{pmatrix} x_1 \\ x_2 \end{pmatrix},$$

and

$$\begin{pmatrix} \Pi_1 \\ \Pi_2 \end{pmatrix} = \begin{pmatrix} \cos(\theta) & -\sin(\theta) \\ \sin(\theta) & \cos(\theta) \end{pmatrix} \begin{pmatrix} p_1 \\ p_2 \end{pmatrix}.$$

With this, the effective Hamiltonian in terms of the normal modes is

$$H_{\text{eff}} = \sum_{j=1}^{2} \left[ \frac{1}{2} m \left( \Omega_j^2 - \widetilde{\Omega}_j^2 \right) \left( \cos^2(\theta) Q_1^2 + \sin^2(\theta) Q_2^2 + 2\cos(\theta) Q_1 Q_2 \right) \right]$$
$$- 2K \left( -\cos(\theta)\sin(\theta) \left( Q_1^2 + Q_2^2 \right) + \cos^2(\theta)\sin^2(\theta) Q_1 Q_2 \right) + \frac{\Pi_1^2}{2m} + \frac{\Pi_2^2}{2m}. \qquad (109)$$

Here we made the assumption of $m_1 = m_2 = m$. To diagonalize the Hamiltonian, the condition on the rotation that should be performed reads as

$$\tan(2\theta) = \frac{2K}{\left( \Omega_1^2 - \widetilde{\Omega}_1^2 \right) - \left( \Omega_2^2 - \widetilde{\Omega}_2^2 \right)}.$$

For this rotation the Hamiltonian is

$$H_{\text{eff}} = \frac{1}{2} m \sum_{\substack{j,j'=1 \\ j' \neq j}}^{2} \left[ \left( \cos^2(\theta) \left( \Omega_j^2 - \widetilde{\Omega}_j^2 \right) + \sin^2(\theta) \left( \Omega_{j'}^2 - \widetilde{\Omega}_{j'}^2 \right) + K\sin(2\theta) \right) Q_j^2 \right] + \frac{\Pi_1^2 + \Pi_2^2}{2m}.$$

$$(110)$$

Hence the system has decoupled into two harmonic oscillators with frequencies

$$\nu_1^2 = \cos^2(\theta) \left( \Omega_1^2 - \widetilde{\Omega}_1^2 \right) + \sin^2(\theta) \left( \Omega_2^2 - \widetilde{\Omega}_2^2 \right) + K\sin(2\theta),$$
$$\nu_2^2 = \sin^2(\theta) \left( \Omega_2^2 - \widetilde{\Omega}_2^2 \right) + \cos^2(\theta) \left( \Omega_1^2 - \widetilde{\Omega}_1^2 \right) - K\sin(2\theta),$$

and with the following second order moments

$$\left\langle Q_j^2 \right\rangle = \frac{\hbar}{2m\nu_j} \left( 2\left\langle n_j \right\rangle + 1 \right) = \frac{\hbar}{m\nu_j} \coth\left( \frac{\nu_j}{2T} \right),$$
$$\left\langle \Pi_j^2 \right\rangle = \frac{\hbar m\nu_j}{2} \left( 2\left\langle n_j \right\rangle + 1 \right) = \hbar m\nu_j \coth\left( \frac{\nu_j}{2T} \right),$$

where the average is a statistical average over the bath variables. Also we find $\left\langle Q_j Q_q \right\rangle = \left\langle \Pi_j \Pi_q \right\rangle = 0$ for $j \neq q$. In the high temperature limit, $\coth\left( \nu_j \hbar / 2k_B T \right) \sim \frac{2Tk_B}{\hbar \nu_j}$ and therefore

$$\left\langle Q_j^2 \right\rangle \sim \frac{2Tk_B}{m\nu_j^2},$$
$$\left\langle \Pi_j^2 \right\rangle \sim 2mTk_B.$$

One can now express the coherences or off-diagonal correlation functions at thermal equilibrium for the initial set of variables as

$$\langle x_1 x_2 \rangle = \sin(\theta)\cos(\theta) \left( \left\langle Q_1^2 \right\rangle - \left\langle Q_2^2 \right\rangle \right). \qquad (111)$$

In case that $\nu_1 \neq \nu_2$, which happens when $\Omega_1 \neq \Omega_2$ and/or $g_k^{(1)} \neq g_k^{(2)}$ for some $k$, then the coherence between $x_1$ and $x_2$ does not cancel out in the high temperature limit. Note however that this does not imply that entanglement survives at the high temperature limit, as this correlation is not necessarily a quantum correlation. This behaviour of $\langle x_1 x_2 \rangle$ is consistently verified in the numerical studies that we undertook using the original Hamiltonian. We note

here that such behaviour, i.e. of asymptotic non–vanishing coherences at the high-temperature limit were identified in [87] but for the case of each particle attached to its own environment and both environments having an ohmic spectral density. The reason for the resemblance of the two cases is that, in our case, one could argue that even though the two particles are coupled to a common bath contrary to [87], the particles effectively see the bath at different temperatures when $\nu_1 \neq \nu_2$. In particular, each one sees the bath at a temperature $\frac{T}{\nu_j^2}$. Hence our system in this case also reaches a non-equilibrium stationary state.

From these considerations, an estimate of the critical temperature can also be obtained. Notice that an important difference between the Hamiltonian in Eq. (110) and the original linear Hamiltonian in Eq. (18) is that the former has a spectrum that is bounded from below. In this case, and as the Hamiltonian is also self-adjoint, one can apply the results of Ref. [88], where it is proven that the critical temperature for such a symmetric system as the one we are considering is given by

$$(k_B T_{\text{crit}})^{-1} = \frac{1}{\hbar \nu_{\max}} \sigma(r), \tag{112}$$

where $\nu_{\max} = \max[\{\nu_j\}]$ with $j \in 1, 2$, $r = \nu_{\max}/\nu_{\min}$ where $\nu_{\min} = \min[\{\nu_j\}]$ and $\sigma(r) = ts(t)$ with $1 \leq t \leq r$ such that $s(t) = s(\frac{t}{r})$ with

$$s(x) := \frac{1}{x} \ln \left| \frac{1+x}{1-x} \right|. \tag{113}$$

For the general values of the parameters given in Sec. 3.2, the value of the critical temperature obtained assuming this effective Hamiltonian, was of the order of $nK$ in agreement with our findings.

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
