# Peer review of "Two distinguishable impurities in BEC: squeezing and entanglement of two Bose polarons"

_SciPost Physics, doi:SciPost Phys. 6, 010 (2019)_

## Round 2 · Referee Report · Anonymous (Referee 1) · 2018-6-24

Strengths

1) addresses challenging problem of quantifying entanglement in a many-body setting.
2) thorough discussion of method is provided.

Weaknesses

1) more details concerning possible experimental verifications could be provided.

Report

In their manuscript, Charalambous and co-workes study entanglement and squeezing properties of two impurity atoms inside a BEC, using a quantum Brownian motion approach. Their work is motivated by the quest to utilize bath-induced interactions to enable quantum information protocols, and to understand the role of the surrounding bath in destroying the mutual entanglement which is vital for such protocols to work efficiently. The authors provide a detailed explanation of their method and a thorough discussion of their results. In my view this work certainly deserves publication, and I recommend accepting it in SciPost.

Requested changes

Before publication, the authors should take into account the following comments:

1) In the discussion of their results, the authors should provide a more concise description of the physical situation under consideration. In particular the precise initial conditions of their non-equilibrium dynamics remained unclear to me. On the bottom of page 18 the authors state that they start from a state without entanglement. However, in Fig. 1, they present results showing how there is entanglement present at intermediate times, which disappears. How does the entanglement evolve at shorter times? Is there some entanglement present when the dynamics are started?

2) The authors should provide some discussion how the quantities they calculate can be measured experimentally. For example, the entanglement is characterized by Eq. (73), where \nu_- is determined as “the smallest symplectic eigenvalue of the partial transpose covariance matrix C”. How can one, at least in principle, measure such a quantity? Similar, abstract properties of the covariance matrix are required in Eq. (79) to characterize squeezing in the system.

3) In the beginning of Sec. 4, the authors introduce a condition where quantum effects play a role, which relies on a comparison of the temperature T to the trapping frequencies. I find this confusing, since I would expect quantum effects to play a role even in a homogeneous system when \Omega = 0 vanishes.

  • validity: high
  • significance: good
  • originality: good
  • clarity: good
  • formatting: excellent
  • grammar: excellent

Author:  Christos Charalambous  on 2018-11-05  [id 336]

(in reply to Report 1 on 2018-06-24)

We have attached a file with our replies to the comments made in the report

Attachment:

Response-to-first-referee.pdf

Anonymous on 2018-11-14  [id 341]

(in reply to Christos Charalambous on 2018-11-05 [id 336])
Category:
remark

In their replies to both referee reports, the authors have addressed all criticism. They have amended their manuscript accordingly. As I mentioned in my first report, I recommend publication of this manuscript in SciPost in its present form and without any further delay. I apologize for my late response to the reviewers reply.

---

## Round 2 · Referee Report · Anonymous (Referee 2) · 2018-9-6

Strengths

1- Careful, clear, and systematic treatment of the system.
2- Reveals the competition between interactions and thermal effects in determining squeezing and entanglement.

Weaknesses

1- Long. Some extra summary needed to make paper accessible.
2- Discussion of validity conditions needs clarifying.
3- A number of small formatting problems to address.

Report

The paper gives a theoretical treatment of two Bose polarons, with a focus on squeezing and entanglement induced by the bath interactions and the role of thermal noise in enhancing and degrading these quantum properties. A thorough analysis including non-Markovian effects is given, and the role of interaction strength and temperature is investigated to find where squeezing and entanglement are optimised. In general the work is interesting, valid and well executed, and the discussion and conclusions are well supported by the results.

I would like to recommend acceptance, however the requested changes should first be addressed, in particular, there is an assumption that the bath operators are in a thermal distribution at specific temperature [(59), (60)]. However, as they consist of Bogoliubov modes, the impurity dynamics must modify the bath oscillators, presumably included in (58) at some level of approximation. What condition does this assumption place on evolution times? This consideration is relevant to discussion after (15).

In such a lengthy paper, I would like to see a summary of approximations and validity conditions, with parameters satisfying them, at the point where the particular results (i.e. figure 1) are first introduced. Some discussion around the validity of the thermal bath approximation would be particularly useful there.

Requested changes

1- page 4 and equations (2*) contain a lot of different uses of the symbol $V$, including as trapping potential, interaction potential, and system volume. Please disambiguate.

2- Make (4) consistent with (2b), (2c) containing $\int d^d\mathbf{x}$ for volume integral.

3- At present it is not entirely clear whether $x_j$, $p_j$ are classical or quantum variables until later in the article. To clarify equations (2*), it would help the reader for commutators to be supplied.

4- In the development of (8), the authors should state that, as usual, the diagonal form is only valid up to quadratic order in the bath operators.

5- The exponents in (12), (14), (16)-(19), etc, should be formatted as a dot product.

6- Equation (20e) is an empty line.

7- In the unnumbered solutions (number these) after (20e), the first term should read $b_k(t_0)e^{-i\omega_k(t-t_0)}$, etc, to be consistent with the initial condition. Presumably $t_0$ should appear in (22), (23) also.

8- Eq. (27) as written is unclear since $B^T$ and $W$ have different dimension, and can't strictly be subtracted.

9- After (36), "In the following sections we solve Eqs (27) and (32) for the cases under study". At this point it would be help the reader (in a long article such as this) to clarify how they are solved.

10- Eq (37) - where does this come from? Presumably the frequencies are found by unpacking (36)?

11- Eq (39) and (40) have different arguments of $\Theta()$, please make consistent.

12- Just before (49): "This can be achieved by introducing an ultraviolet cutoff given by $\Lambda$ such that only the part of the spectrum where $\omega\ll \Lambda$ remains." is not consistent with the way the cutoff is implemented. It should read $\omega <\Lambda$. [I do not refer to whether the physics is dominated by $\omega\ll\Lambda$, a separate issue, as clarified after (51)]. Both implementations (50) and (51) retain $\omega <\Lambda$.

13- $\mathcal{E}_{LN}$ introduced in (73) is different from the y-axis label used in the figures. Please make these consistent, and provide a reference from the caption of figure 1 to (73), so that the more casual reader may identify the definition without combing the text.

14- Appendix B: the imaginary part is italic but real is roman. Please make consistent.

15- Appendix B: before (96) a definition of the incomplete gamma function is given, however there is an upper and a lower, and this appears to be an unusual definition. Please specify.

16- Provide a summary of validity conditions for assumptions, and parameters satisfying them at the end of the formalism, or start of results section.

  • validity: high
  • significance: good
  • originality: good
  • clarity: good
  • formatting: good
  • grammar: reasonable

Author:  Christos Charalambous  on 2018-11-05  [id 335]

(in reply to Report 2 on 2018-09-06)

We have attached a file with our replies to the comments made in the report

Attachment:

Response-to-second-referee.pdf

Anonymous on 2018-12-18  [id 389]

(in reply to Christos Charalambous on 2018-11-05 [id 335])

The authors have addressed all of my comments in an entirely convincing way, and I therefore recommend publication without further delay.

---

## Round 3 · Referee Report · Anonymous (Referee 3) · 2018-12-8

Report

In their replies to both referee reports, the authors have addressed all criticism. They have amended their manuscript accordingly. As I mentioned in my first report, I recommend publication of this manuscript in SciPost in its present form and without any further delay. I apologize for my late response to the reviewers reply.

---

## Round 3 · Referee Report · Anonymous (Referee 4) · 2018-12-18

Report

The authors have addressed my comments in an entirely convincing way. I recommend publication of the manuscript without further delay.

---

## Editorial Decision

published